

# Satellite inference of water vapor and aerosol-above-cloud combined effect on radiative budget and cloud top processes in the Southeast Atlantic Ocean

Lucia T. Deaconu[1,2], Nicolas Ferlay[2], Fabien Waquet[2], Fanny Peers[3], François Thieuleux[2], Philippe Goloub[2]

[1] Department of Physics, University of Oxford, OX1 3PU, Oxford, UK
[2] Université de Lille, CNRS, UMR 8518, LOA – Laboratoire d'Optique Atmosphérique, 59000 Lille, France
[3] College of Engineering, Mathematics and Physical Sciences, University of Exeter, Exeter, UK

*Correspondence to*: Lucia T. Deaconu (lucia.deaconu@physics.ox.ac.uk) and Nicolas Ferlay (nicolas.ferlay@univ-lille.fr)

**Abstract.** Aerosols have a direct effect on the Earth's radiative budget and can also affect the cloud development and lifetime, and the aerosols above clouds (AAC) are particularly associated with high uncertainties in global climate models. Therefore, it is prerequisite to improve the description and understanding of these situations. During the austral winter, large loadings of biomass burning aerosols originating from fires in the southern African subcontinent are lifted and transported westwards, across the Southeast Atlantic Ocean. The negligible wet scavenging of these absorbing aerosols leads to a near-persistent smoke layer above one of the largest stratocumulus cloud deck on the planet. Therefore, the Southeast Atlantic region is a very important area for studying the impact of above cloud absorbing aerosols, their radiative forcing and their possible effects on clouds.

In this study we aim to analyse and quantify the effect of smoke loadings on cloud properties using a synergy of different remote sensing technics from A-Train retrievals (methods based on the passive instruments POLDER and MODIS and the operational method of the spaceborne lidar CALIOP), collocated with ERA-Interim re-analysis meteorological profiles. To analyse the possible mechanisms of AACs effects on cloud properties, we developed a *high* and *low* aerosol loading approach, that consists in evaluating the change in radiative quantities (i.e. cloud top cooling, heating rate vertical profiles) and cloud properties with the smoke loading. During this analysis, we account for the variation in the meteorological conditions over our sample area.

The results show that the region we focus on is primarily under the energetic influence of absorbing aerosols, leading to a significant positive shortwave direct effect at the top of the atmosphere. For larger loads of AACs, clouds are optically thicker, with an increase in liquid water path of 20 g.m$^{-2}$ and lower cloud top altitudes by 100 m. These results do not contradict the semi-direct effect of above cloud aerosols, explored in previous studies. Furthermore, we observe a strong correlation between the aerosol and the water vapor loadings, which has to be accounted for. A detailed analysis of the heating rates shows that the absorbing aerosols are 90 % responsible for warming the ambient air where they reside, with approximately +5.7 K/day, while the accompanying water vapour above clouds has a longwave effect of +4.7 K/day (equivalent to 7% decrease) on the cloud top cooling. We infer that this decreased cloud top cooling in particular, in addition with the higher humidity above the clouds, might modify the cloud top entrainment rate and its effect, leading to thicker clouds. Therefore, smoke (the



combination of aerosol and water vapor) events would have the potential to modify and probably reinforce the underlaying cloud cover.

## 1    Introduction

The South Atlantic Ocean (SAO) is covered almost permanently by the largest stratocumulus cloud deck on the planet.
These clouds play a very important role in the climate system, as they cool the tropics by reflecting sun radiation back into space (Bretherton et al., 2004; Wood, 2012). Between June and October, biomass-burning aerosols originating from fires in the southern African subcontinent are lifted and transported long distances westwards, mostly above the low-level clouds (Ichoku et al., 2003; Waquet et al., 2009). The southern African subcontinent is the main annual contributor of biomass burning aerosols, as most of this region is covered by savannah. Due to agricultural practices, the savannah vegetation is burned and a large amount of aerosol is injected into the atmosphere (Labonne et al., 2007). The negligible wet scavenging of the aerosols transported above the clouds leads to a near-persistent smoke layer above the stratocumulus deck, that can be suspended in the atmosphere for several days. These dark-coloured aerosols are efficient in absorbing shortwave radiation, which can warm the lower troposphere and can modify the radiative budget, with a large-scale effect on climate that is not yet well understood. For instance, at the top of the atmosphere, the sign and amplitude of the direct radiative effect (DRE) depends on the aerosol properties and on the reflective properties of the underlying surface. The aerosol DRE can be positive or negative, depending on the cloud albedo, the aerosol type and its level of absorption (Lenoble et al., 1982; Keil and Haywood, 2003; Peers et al., 2015). The aerosol effects are, in fact, even more complex and may stabilize, promote or suppress the cloud formation, as a function of the position of the absorbing aerosol layer with respect to the cloud layer and the contact with the cloud droplets. In case of absorbing aerosols above stratocumulus clouds, the warming of the layers located above the clouds would tend to stabilize the boundary layer. This stabilization would induce a lower entrainment rate and a moister boundary layer. The resulting effects would be an increase of the liquid water content and preservation of cloud cover (Brioude et al., 2009; Johnson et al., 2004). On the contrary, if the aerosols are within the cloud, the warming of the atmospheric layer due to aerosol absorption tends to reduce the relative humidity and the liquid water content, decreasing the stratocumulus cloud cover (Hill et al., 2008). Additionally, an increase in the number of aerosols serving as cloud condensation nuclei (CCN) could lead to a larger number of smaller cloud droplets that bring about more reflective clouds, inducing a cooling of the Earth-Atmosphere system (Twomey, 2007). The reduction of cloud droplet size may potentially have other impacts on precipitation and cloud properties.

All these considered, the Southeast Atlantic region is well suited to investigate the interactions of aerosols above clouds (AAC) with radiation and clouds. Currently the complexity of these interactions in the Southeast Atlantic and their influence on local and global climate are, however, not well captured by models. This is mainly due to limitations in accurately representing the aerosol and cloud spatial and vertical distributions, the aerosol absorption capacity and the cloud properties, such as cloud optical thickness, liquid water path and effective radius. In-situ measurements can be used to improve our



understanding of the aerosol direct radiative forcing and cloud adjustments to the presence of aerosols. Recent airborne campaigns aim to answer several scientific questions related to the smoke-and-cloud regime over the Atlantic region using multiple aircraft and surface-based instrumentation located at different measurement sites, over the period 2016 to 2018 (Zuidema et al., 2016). The three campaigns (ORACLES, CLARIFY and AEROCLO-sA) offer the opportunity for

international scientists to collaboratively verify and validate different satellite measurements and constrain the climate models.

Although in-situ measurements are more detailed and provide a better characterization of aerosols and their effects, these local measurements are not sufficient for regional or global climate studies. Space borne observations allow retrieving aerosol and cloud properties over large spatial-temporal scale, facilitating the study of their effects on climate and reducing the related uncertainties. The A-Train satellite constellation includes different passive and active sensors that provide near-

simultaneous measurements of aerosol and cloud properties, allowing the combination and comparison of various methods and also the possibility to perform instrumental synergies. The study of the aerosol above clouds properties and radiative impacts as well as their potential interactions with the underlying clouds using satellite observations is a relatively recent topic in the field of remote sensing. Until now, different methods and instruments were used for the retrieval of the aerosol above clouds (AAC) properties. Lidar instruments that provide the vertical profile of the atmosphere are valuable tools for the study

of aerosol above cloud scenes. The lidar CALIOP (Cloud-Aerosol Lidar with Orthogonal Polarization), installed on the CALIPSO (Cloud-Aerosol Lidar and Infrared Pathfinder Satellite Observation) satellite, uses backscatter measurements to determine the vertical structure of the atmosphere and the properties of the aerosol and cloud layer (Vaughan et al., 2009; Winker et al., 2009; Young and Vaughan, 2009). The operational method developed for CALIOP allows the retrieval of the aerosol properties (i.e. mainly the aerosol optical thickness, AOT) for scenes with aerosols above clouds. However, the method

relies on assumptions and alternative CALIOP-based research methods have also been introduced to retrieve the above-cloud AOT (ACAOT). The depolarization ratio method (Hu et al., 2007) and the colour ratio method (Chand et al., 2008) use fewer assumptions for the retrieval of ACC properties. These techniques are based on light transmission methods and treat the liquid water clouds situated underneath the aerosol layer as a target. Passive sensors have also been used to obtain information on aerosols above clouds. The multidirectional polarization measurements have shown sensitivity to scenes with aerosols above

clouds (Waquet et al., 2009, Hasekamp, 2010; Knobelspiesse et al., 2011), as they strongly modify the polarized light reflected back to space by the cloud layer. Waquet et al. (2013) have developed an operational method for retrieving the properties of AACs that relies on the polarized radiances measured by POLDER (Polarization and Directionality of Earth Reflectances) instrument, on-board of the PARASOL (Polarization and Anisotropy of Reflectances for Atmospheric Science coupled with Observations from a Lidar) satellite. The method is able to retrieve the AOT at 865 nm and the Ångström exponent, which is

a parameter indicative of the particles size. Furthermore, Peers et al. (2015) have developed a complementary method that uses additional total multidirectional radiances measured by POLDER. The method provides the aerosol single scattering albedo (SSA) and the cloud optical thickness (COT). Passive sensor technics that solely use total radiance measurements have also been used to obtain information on aerosols above clouds. Torres et al. (2012) have developed an algorithm to retrieve the ACAOT and the underlying COT, using radiance measurements performed in the ultra violet (UV) by the with Ozone



Monitoring Instrument (OMI) instrument. Similar methods that can retrieve the above cloud AOT and, simultaneously, the cloud properties have been also developed for the Moderate Resolution Imaging Spectroradiometer (MODIS) instrument (Jethva et al., 2013, Meyer et al. 2015).

Previous studies that aimed to analyse the impact of absorbing aerosols on the cloud properties and radiative forcing were based on the exploitation of several A-Train satellite observations and modelling. Costantino and Bréon (2013) used MODIS to retrieve aerosol and cloud properties, collocated with CALIOP estimates of aerosol and cloud altitudes. Their objective was to use the simultaneous satellite retrieved aerosol and cloud properties to contribute to the knowledge of aerosol effect on low-level stratocumulus cloud microphysics (droplet effective radius, $r_{eff}$), optical properties (COT) and liquid water path (LWP). Their results showed that the aerosol effects on the cloud microphysics are strong when the layers are in contact: effective radius can decrease from 15-16 μm down to 10-11 μm for an aerosol index that varies from 0.02 to 0.5, suggesting a potential indirect effect of aerosols. Wilcox (2010) also used the aerosol and cloud altitudes retrieved with CALIOP in combination with OMI aerosol index, in order to analyse the link between the absorbing aerosols located above clouds and the marine stratocumulus cloud properties. His results showed that the presence of absorbing aerosol layers lead to a heating (by nearly 1 K at 700 hPa) in the lower troposphere that stabilizes the atmosphere. This warming coincides with LWP values greater by more than 20 g.m$^{-2}$ and cloud top altitude lower by 200 m in cases when high loads of smoke are transported above the cloud. Sakaeda et al. (2011) obtained similar results using the large-eddy Community Atmospheric Model 3.0 (CAM) constrained by satellite observations.

As mentioned above, the effect of aerosols on the cloud cover can be complex and indirect and it can result from the modification of the atmospheric thermodynamics by the aerosol loading, e.g. a modification of atmospheric stability. But to go further, a difficulty and challenge in the analysis of aerosol effects on clouds comes from the fact that cloud properties are also, and often primarily, sensitive to meteorological conditions and corresponding atmospheric thermodynamics and dynamics. Studies dedicated to aerosol effects should thus aim to disentangle, if possible, aerosol effects on cloud properties from effects of meteorological conditions only (Brenguier and Wood, 2009; Stevens and Brenguier, 2009). A level of complexity comes from the fact that covariances exist between meteorological parameters and aerosol concentration and properties. Climatology of biomass-burning events may coincide with changes in meteorological regimes. For example, Adebiyi et al. (2015) showed a shift southward in circulation patterns and thermodynamics between July-August and September-October, as the southern African anticyclone strengthens. Also, while the aforementioned studies depict dry smoke plumes, biomass-burning aerosol events can be accompanied by varying water vapour production. Depending on the moisture content of fresh biomass (Parmar et al., 2008), the natural or anthropogenic biomass fires are indeed releasing water vapour in the atmosphere, in addition to organic and black carbon, $CO_2$ and CO (Levine, 1990). It might be important to account for the effect of this accompanying moisture, and to identify the different air circulation patterns that will lead the biomass-burning transportation off coast of South Africa. It was done by Adebiyi et al. (2015) who incorporated radiosondes measurements from St. Helena Island of temperature and specific humidity, MODIS AOT$_{550nm}$, CALIOP aerosol altitude data and reanalysis data (ERA-Interim) to provide a unique dataset of thermodynamic profiles linked to clear and polluted conditions. They also



investigated the radiative effect of moisture and absorbing aerosol in different cloudy conditions at St. Helena. They show that the specific humidity ($q_v$) is higher within the aerosol plumes (around 700 hPa). This was previously observed during the UK-SAFARI 2000 campaign (Haywood et al., 2003) with $q_v$ values larger than 2 to 4 g.kg$^{-1}$ within the aerosol layer, while outside the smoke plume the $q_v$ values are less than 1 g.kg$^{-1}$. The moisture may have a role in the aerosol aging (Dubovik et al., 2002; Haywood et al., 2003; Kar et al., 2018) and a radiative significance in both shortwave and longwave spectra. Radiative transfer calculations show that mid-tropospheric moisture generates a net diurnal cooling of approximately 0.45 K/day, decreasing the impact of the shortwave heating caused by the biomass-burning aerosols that reaches 1.5 K/day. As in Wilcox (2010), this study shows a decrease in the cloud top altitude of about 112 m near St. Helena under polluted conditions.

Starting from all these studies, we have developed our own strategy in order to better understand the effect of aerosols and meteorological parameters on clouds. Our strategy exploits several satellite measurements and meteorological reanalysis data as constraints of a radiative transfer model. We focus on situations with large amounts of absorbing aerosols above the clouds and compare them with cases characterized by low values of absorbing AOT. Also, in order to minimize the meteorological impact, we analysed only a limited time period over a smaller area off coast of Angola. Smoke layer elevated above the clouds can induce a semi-direct effect through an energetic forcing due to both the effects of biomass burning aerosol and water vapour. We therefore developed a method to calculate the profiles of heating rates in the visible and thermal infrared by combining POLDER and CALIOP data as well as meteorological data. In section 2, we present the different parameters used to estimate the AAC, cloud properties and the meteorological variables and some elements of climatology available for the South East Atlantic Ocean. We also describe the CALIOP / POLDER synergy and the radiative transfer calculations. Section 3 describes a "*low* and *high* approach", which consists in distinguishing between two different aerosol-loading situations, and analysing the difference in the statistic of cloud properties and meteorological parameters. We present this strategy in terms of selected area, time period and covariance between aerosol load and meteorological data. Section 4 shows the changes in cloud properties with respect to the aerosol loadings, the overall radiative effect of the smoke layer for the selected region and an attempt to separate the radiative effects of the biomass-burning aerosols from those of the water vapour. Section 5 provides the conclusions and perspectives.

## 2   Dataset and methodology

### 2.1   Description of the dataset

In this study, we use the version 4.00 of the official output product PARASOL_PM02-L2 for AAC scenes (available at ICARE website, http://www.icare.univ-lille1.fr/parasol/products/) to retrieve the properties of aerosols above clouds. The parameters used in our study are the aerosol optical thickness, the aerosol single scattering albedo and the aerosol-corrected cloud optical thickness. All these parameters are provided at a resolution of $6 \times 6$ km$^2$ and available between 490 and 865 nm. We also use the Ångström exponent, which is primarily indicative of the particles size (i.e. with the AOT retrieved at 865 and 670 nm). The aerosol model prescribed by the algorithm (i.e. particles size and absorption) is also used to extrapolate these



optical properties over a large spectral range. The uncertainties associated with these variables are thoroughly described in Peers et al. (2015b) and Waquet et al. (2013b). The lidar CALIOP is primarily used to determine the aerosol layer vertical extent. We used the level 2 version 3.01 of the inversion products, officially named CAL_LID_L2_05kmALay for the aerosol base and top altitudes, CAL_LID_L2_05kmCLay for the cloud top altitude and CAL_LID_L2_05km_APro for the vertical

profile of aerosol extinction at 532 nm (data can be found at http://www.icare.univ-lille1.fr/calipso/ products/) (Vaughan et al., 2009; Young and Vaughan, 2009). The other properties that we analyse for the clouds are the droplets effective radius, $r_{eff}$ inferred by MODIS, the liquid water path, LWP, and the cloud altitude, $ZO_2$ derived from POLDER. The cloud altitude is calculated using the POLDER oxygen pressure method ($P_{Oxygen}$), which is determined from differential absorption between two spectral bands centred on the oxygen A-band (763 and 765 nm respectively) (Buriez et al., 1997; Vanbauce et al., 2003).

In our study, the $ZO_2$ product is used as proxy for the cloud top altitude estimate outside the CALIOP track. The liquid water path is computed with the homogeneous assumption, using the aerosol-corrected cloud optical thickness from PARASOL and the MODIS droplets effective radius:

$$LWP = \frac{2\rho_w}{3} \times COT \times r_{eff}$$
(Eq. 1)

where $\rho_w$ is the water density. The MODIS $r_{eff}$ bias for biomass-burning aerosols above clouds (Meyer et al., 2015), is expected to be of about 2% on average and then should have a small effect on the LWP computation.

In order to estimate the main thermodynamic quantities of the atmosphere, we use the ERA-Interim product provided by the European Centre for Medium-Range Weather Forecast (ECMWF) model (Berrisford et al., 2011). This reanalysis meteorological database delivers various parameters, including profiles of temperature (K), specific humidity (g.kg$^{-1}$), pressure

(hPa), wind speed (m.s$^{-1}$), wind direction (°) and ozone (atm-cm). The assimilating model is configured for 60 vertical levels, from the surface up to 0.1 hPa. The horizontal resolution of the products is 0.5 degree and the reanalysis data are provided every 6 hours (Dee et al., 2011). In our study, these data were spatially and temporally collocated with the POLDER and CALIOP observations.

**2.2    Radiative transfer calculation and synergy CALIOP / POLDER**

To compute fluxes and heating rates at different levels in the atmosphere, in the visible and also in the thermal infrared, we use the Global Atmospheric ModEl (GAME), which is a fast and high spectral resolution radiative transfer code that allows the calculations of radiances (and fluxes) in horizontal and plan-parallel atmospheric layers (Dubuisson, P., J. C. Roger, M. Mallet, 2006). The model accounts for the Rayleigh scattering effects and for the scattering and absorbing properties of clouds and aerosols that have to be computed over the different spectral intervals: 208 spectral intervals for the shortwave spectrum

(from 220 nm to 4 µm) and 115 spectral intervals for the longwave spectrum (from 4 to 50 µm). This method allows discretizing



the radiation field in a finite number of propagation directions and allows to efficiently compute the multiple scattering processes occurring in the atmosphere whatever the value of the optical thickness.

GAME model requires information on the vertical distribution of aerosol and clouds, their optical and microphysical properties and the vertical structure of the atmosphere. For this we use the collocated POLDER, CALIOP and ERA-Interim re-analysis data for the temperature and humidity profiles, which are crucial for the radiative computation in the longwave spectrum.

In GAME the computation is made for plane-parallel layers of atmosphere, each layer characterized by particular values of aerosol (or cloud) and meteorological parameters. We have divided the lower troposphere into 100 m-thick layers from the surface up to 8 km. Above this altitude (up to 100 km) the layers are more roughly distributed, as the upper atmosphere is not under the influence of biomass-burning aerosols and less our field of interest. For each 100 m-aerosol layer we provided the CALIOP extinction coefficient ($\sigma_e$, km$^{-1}$.sr$^{-1}$).

However, the CALIOP method can underestimate the geometrical thickness of the aerosol layer when the optical thickness is large enough, due to the attenuation of the backscattered signal (Jethva et al., 2014). The CALIOP miscalculation of the aerosol bottom altitude would implicitly underestimate the aerosol extinction profile (i.e. the AOT), hence the aerosol radiative forcing. A recent study that uses independent lidar backscattering measurements at 1064 nm of the Cloud-Aerosol Transport System (CATS) (Yorks et al., 2014), shows that the CALIOP algorithm probably overestimates the base of the aerosol layer by 500 m (Rajapakshe et al., 2017). Deaconu et al. (2017) showed that the CALIOP operational algorithm underestimates the AOT above clouds with a factor of 2 to 4 depending on the aerosol type, when compared to other methods dedicated for aerosol above cloud retrievals - the POLDER polarisation method (Waquet et al., 2009) and the CALIOP depolarisation ratio method DRM (Deaconu et al., 2017; Hu et al., 2007). By analysing the consistency between the aerosol above cloud retrievals from POLDER method and CALIOP DRM, they also found good agreement for AOT retrievals when the microphysics of the aerosols is well defined (such as biomass-burning aerosols) and when the aerosol layer is detached from the cloud. These results give confidence in our ability to measure the properties of AAC over the South Atlantic region using POLDER method. Therefore, in our study we will use the POLDER AOT$_{865nm}$ retrieved under the CALIOP track to scale the CALIOP $\sigma_e$ profile used in GAME (Figure 1). As the POLDER AOT is retrieved at 865 nm and the CALIOP $\sigma_e$ is at 532 nm, the $\sigma_e$ scaling firstly requires an extrapolation of POLDER AOT at 532 nm. Afterwards, we infer the resulted CALIOP $\sigma_{e,scaled}$ at 550 nm, which is the native wavelength used for AOT in the radiative transfer model.

We compute the instantaneous heating rate (HR) profiles (K/day) in the shortwave and in the longwave domains (HR$_{SW}$ and HR$_{LW}$). The first is due to the shortwave absorption of aerosols and water vapour. The second is the result of infrared absorption and emission processes, and depends mainly on the profile of atmospheric component, cloud and water vapour, with their temperatures. In the longwave spectrum we considered a negligible effect of aerosols. The sum between HR$_{SW}$ and HR$_{LW}$ gives the total heating rate, referred as HR$_{total}$ in the following.



### 2.3 Elements of climatology in the South East Atlantic Ocean

### 2.3.1 Meteorological patterns

The pattern of the atmospheric circulation plays a determinant role in the transport of aerosols emitted from the African continent. Meteorology and circulation patterns can also impact the stratocumulus clouds by changing the
thermodynamic environment, regardless of the presence (or absence) of aerosols above clouds. Therefore, it is necessary to separate (or constrain) the effects of meteorology on clouds from the variations caused by AAC in order to study the aerosol effects on clouds in the SAO. In this area, the main atmospheric circulation is dominated by easterlies throughout the air column and south-easterlies close to the surface, as shown in the study of Adebiyi et al. (2015) figures 8 to 10. From July to October the southern hemisphere is influenced by the Atlantic anticyclone between 600 hPa and 800 hPa, and the trade winds
at the surface (with winds stronger than 5 m.s$^{-1}$). The September-October period presents differences compared to the July-August period, as different circulation patterns cause a maximum coverage of the stratocumulus clouds, and more importantly, a maximum in the continental aerosol loading transported westwards over the Atlantic basin, around 10°S. This region is also characterized by two different humidity and temperature regimes: larger values are found north of 20°S compared to the southern region. Adebiyi et al. (2015) also described a general increase in the mid-tropospheric moisture at 600 hPa during
September-October, suggesting an association between the aerosol loading and moisture.

### 2.3.2 Aerosol and cloud properties

The maps shown in Figure 2 present the average values of different POLDER and MODIS aerosol and cloud retrievals at a horizontal resolution of 6×6 km$^2$, acquired from May to October 2006 to 2009 over an area that extends from 30°S to 5°N and from 12°W to 14°E in the SAO.
Large aerosol loadings are found close to the coast, where the average above-cloud AOT exceeds 0.3 at 865 nm. Towards the west, the ACAOT decreases to an average of 0.2 at 865 nm due to transportation and deposition of aerosols (Figure 2a). As expected, the mean values of the Ångström exponent (AE$_{670/865}$) along the shoreline are larger than 2.0, characteristic of very fine particles of smoke (Dubovik et al., 2002), while westwards the mean AE$_{670/865}$ values slightly decrease to around 1.85 (Figure 2b). This suggests an increase in the particle size, as the plume is transported towards the open
sea. The decrease of the mean AE values with longitude can suggest the evolution of the aerosol properties, such as growth of the smoke aerosols associated to aging (Kar et al., 2018; Müller et al., 2007; Waquet et al., 2013; Reid et al., 1998). Also, between 0° and 5° North, the values of the AE$_{670/865}$ go down to 0.4 (not showed on the colour scale), which correspond to the dust particles in the POLDER method, indicating that dust particles could be preferentially transported above clouds over this area.
Information related to the absorption property of the aerosols is given by the single scattering albedo and/or by the absorption AOT$_{865nm}$, which is the product of the extinction (total) AOT by (1 – SSA). For this region we observe high values of absorption AOT$_{865nm}$, which exceed 0.04 close to the coast (Figure 2c) and decrease progressively westwards. These



estimates are consistent with the values calculated for the $SSA_{865nm}$ of 0.80 at seashore that increase to 0.87 around 12°W (Figure 2d). The observations are in agreement with the typical values provided by AERONET (Dubovik et al., 2002; Sayer et al., 2014) and the field campaigns Southern African Regional Science Initiative (SAFARI-2000; Leahy et al., 2007; Swap et al., 2002) and Dust and Biomass Experiment (DABEX; Johnson et al., 2008) for the biomass-burning aerosols. North of the

latitude of 5°S, the SAO region is under the influence of dust particles and the retrieved $SSA_{865nm}$ is up to 0.97, which is expected for mineral dust particles that do not much absorb light in the near-infrared (Dubovik et al., 2002).

Low-level stratocumulus clouds typically cover the South Atlantic Ocean. Generally, these clouds are characterized by rather small droplets ($r_{eff}$ of approximately 10 μm) and by optical thickness values of around 10-15 (Szczodrak et al., 2001). In the presence of above-cloud biomass-burning aerosols, the satellite retrieved COT can be underestimated by more than 20

%, especially over bright clouds with large COT (Haywood et al., 2004; Peers et al., 2015). Figure 2e presents the mean POLDER COT that was corrected for the aerosol induced bias due to aerosol above-cloud absorption (Peers et al., 2015). The MODIS cloud droplet effective radius ($r_{eff}$) (Figure 2f) is increasing from the coast towards the open sea, from 10 to 15 μm. Figure 3 compares the POLDER $ZO_2$ product with the CALIOP cloud top altitude (CTA). In case of geometrically thick clouds, the Oxygen pressure method indicates mainly the cloud middle pressure (Ferlay et al., 2010), instead of cloud top

pressure. This explains the difference that we observed between POLDER $ZO_2$ and the CALIOP (CTA) over a period of six months (May to October) from 2006 to 2010 along the CALIOP track: the two altitude measurements are well correlated but $ZO_2$ is lower that CTA. From the side histograms, we notice that the majority of CTA values are centred around 1.25 km, while $ZO_2$ values are centred around 0.9 km. The difference between the products increases systematically when CTA exceeds 1.5 km. Nevertheless, the stratocumuli are low-level clouds, so, an underestimation of around 300 m by the POLDER product

is more likely. In Figure 2g we observe a minimum cloud altitude of 1.2 km central to the stratocumulus deck, which increases radially as the stratocumulus become more fractioned (i.e. cumulus) or when the frequency of other types of clouds increases (Warren et al., 1988).

Figure 4 shows the variation with longitude of cloud and aerosol altitudes retrieved with CALIOP over the South Atlantic for a period of six months (May to October) from 2006 to 2010. We reported the mean values of cloud top altitude, aerosol

top altitude (ATA) and aerosol base altitude (ABA) for longitude bins of 4°. The data are shown for May-July in the first row and for August-October in the second row. The mean, standard deviation and median, as well as the number of measurements for each 4° bin are provided in the associated tables. For both time periods, we notice the cloud top altitude increasing from 1 to 1.5 km towards the west. This suggests that the clouds develop and become more convective further away from the coast. The average aerosol top altitude is higher during the second interval (August-October), and we observe a higher number of

AAC situations. This is likely due to the fact that the biomass-burning events that reach higher altitudes are most frequent during the late austral winter. We also observe a double layer aerosol profile in the first period, mainly west of 1°E (upper left profiles): one mode around 3.0 km and another around 1 km, which are likely to correspond to sea salt aerosols located in the boundary marine layer. In the first period the aerosol base and top altitudes don't show strong variability across the longitude, while in the second period the average aerosol altitude decreases from east to west. This suggests that contact situations





between the cloud and the aerosol layers are less frequent close to the coast and increase as the aerosols are transported westwards.

## 3    The high and low smoke loading approach

### 3.1    Strategy of analysis

The final objective of the current study is to analyse if there are differences in cloud properties as a function of aerosol loadings, and if these differences can be understood and attributed to the overlying aerosols or to the meteorological variability. Here, we propose a *low* and *high* approach which consists in distinguishing between two different aerosol loading situations, and analysing the difference in the statistic of cloud properties and meteorological parameters, for a selected temporal and spatial domain. In the following, for the *high* cases the above cloud absorption $AOT_{865nm}$ is larger than 0.04, while the *low*

cases are characterized by absorption $AOT_{865nm}$ lower than 0.01. In this section, we first justify the choice of the sample area and time period selected for this study. We then analyse the covariance relationship between the aerosol load and the water vapour content and explore the meteorological patterns for the selected area in function of the *high* and *low* scenario.

### 3.2    Selection of a sample area and time period

We selected a region close to Angola's coast that expands from 15° to 10°S and 6° to 14° E (Figure 2a - black box),

which is close to the main stratocumulus region identified by Klein and Hartmann (1993). The spatial size of the domain is limited in order to control the natural variability of cloud properties and meteorology. Also, the domain is close to the coast so that aerosol loading is high (Figure 2) and aerosols are mainly detached from low level clouds (Figure 4). This is important in order to minimize the microphysical interactions between aerosols and cloud droplets (i.e. indirect effect) and the probability of aerosols within the clouds, which could affect the retrieval of aerosol properties with the POLDER polarisation method

(Deaconu et al., 2017).

The variability of meteorological parameters in this sample area is emphasized in Figure 5, which presents ERA-Interim monthly mean meteorological parameters from June to October 2008. We can observe that the temperature profile doesn't change much throughout this period, while the relative humidity (RH) and the specific humidity ($q_v$) change from month to month. In June and July, moisture levels are comparable, with values of $q_v$ lower than 2.5 g.kg$^{-1}$ at 700 hPa, in contrast

with August and September where $q_v$ is reaching 5 g.kg$^{-1}$ at 700 hPa. In October, RH shows a strong peak between 700 and 500 hPa, and $q_v$ is also larger compared to previous months. The average monthly horizontal winds show a significant difference between the months of SO compared to JJA. The winds are much stronger in SO and much more westwards above 800 hPa. The wind speed also increases above 800 hPa during these months. Winds in August differ from the winds in June/July, but not significantly below 800 hPa.

Knowing these temporal variations of mean meteorological parameters, we chose to analyse the aerosol and cloud parameters and their correlation over the months of June to August 2008. Doing so, we mostly select one meteorological





regime characterized by few differences at the cloud altitudes in terms of wind, temperature and humidity and a moderate water vapour content at the aerosol layer.

### 3.3    Covariance between humidity and aerosol loading

We analysed the mean values of the specific humidity from June to August 2008 over the sampling area, at different pressure levels, as a function of the aerosol loading. Figure 6 shows the ERA-Interim $q_v$ values at 6 pressure levels in function of the POLDER AOT at 865 nm. At the surface and at 950 hPa, the average $q_v$ is almost constant, varying slightly with the AOT from 10 to 12 g.kg$^{-1}$ and from 7.5 and 10 g.kg$^{-1}$, respectively. Higher in altitude, at 500 and 400 hPa the mean values of $q_v$ are very small, regardless of the aerosol loading. On the other hand, at the smoke plume level, at 850 and 700 hPa we notice a strong increase of the $q_v$ with the AOT, from 2 to 7 g.kg$^{-1}$ and from 1 to 5 g.kg$^{-1}$, respectively. So, there is a strong correlation between humidity and aerosol loading at aerosol's altitudes, which confirm previous funding from Adebiyi et al., 2015.

An interpretation of the humidity reinforcement with larger biomass-burning AOT could originate from combustion processes. Depending on the fire intensity and the meteorological conditions, smoke parcels can be saturated with water vapour and the latent heat resulted from the condensation of the water vapour can enhance the vertical development of convection due to additional buoyancy. It is possible that smoke and water vapour released during biomass combustion are both advected at higher altitudes, which contributes to the humidity in the aerosol plumes. Through laboratory measurements and experimental studies scientists have attempted to confirm that water vapour from wild-land or grass fires can significantly modify the dynamic of the lower troposphere. Some of their results suggested that the fuel moisture could make a significant contribution to the humidity within the aerosol plumes resulted from biomass combustion (Clements et al., 2006; Hudspith et al., 2017; Parmar et al., 2008).

### 3.4    Meteorological patterns for high and low situations

We investigate the mean temperature, specific humidity, relative humidity and atmospheric subsidence provided by ERA-Interim, over JJA 2008, for *high* and *low* smoke loading situations. Because of the correlation between aerosol and water vapour, in our study smoke loading means aerosol and water vapour loading.

Figure 7a shows that the temperature profile is almost identical between high and low cases throughout the atmospheric column. In both cases, we notice a temperature inversion, called subsidence inversion, that occurs as result of adiabatic compression when high-pressure systems sink the upper air layers. A moderate to strong temperature inversion almost always caps stratocumulus clouds located under a high-pressure centre (such as the South Atlantic anticyclone). When smoke layers heated by the solar radiation cover low-level stratocumulus clouds, the temperature inversion is strengthened (Kaufman et al., 2005). In our case, we can observe a small increase by 1 K at 850 hPa for *high* smoke loadings.

The average humidity profiles (Figure 7b and 8c) show some significant differences between the two cases. For the *high* cases the average relative humidity (RH) is almost saturated at the cloud level and presents a strong peak at 700 hPa (RH ≈ 40 %). The RH difference between *high* and *low* situations reaches a maximum between 800 and 600 hPa, where the smoke



layer resides. Also, the specific humidity is higher (up to 2.5 g.kg$^{-1}$) everywhere throughout the air column for the polluted cases and we notice an increase of $q_v$ by 300 % at 700 hPa compared with only 35.7 % at 925 hPa (pressure level mostly associated to the cloud level). On the contrary, subsidence is stronger by about 1 Pa/min at 700 hPa when *low* loadings of aerosols are present above the clouds (mainly during June-July). It is expected that the large-scale subsidence decreases when
aerosol loadings are higher (Adebiyi et al., 2015 see Figure 15), which would allow the cloud to rise in altitude.

The meteorological database allows us to compute wind speed roses and specific humidity roses, representative for *high* and *low* aerosol situations. This will give information on the circulation of air masses (e.g. wind direction) that put in motion the humidity and the aerosols close to the coast. Figure 8 presents the frequency distribution of these parameters over the sample area in JJA 2008 period, for two pressure levels: 925 hPa – mainly corresponding to the cloud top altitude (Figure
8a) and 700 hPa – where aerosols are found (Figure 8b). At 925 hPa we notice that both situations are characterized by south-easterlies, with wind speeds of 6 to 9 ms$^{-1}$ for more than 50 % of the cases and with specific humidity that is usually larger than 9 g.kg$^{-1}$. Figure 8b shows that the meteorological parameters at 700 hPa are, however, very different for the two conditions. It is obvious that the air masses carrying *high* loads of smoke are predominantly coming from the land (direction E-NE), while the air circulation responsible for *low* absorption AOT is originating from the open ocean (main direction W-NW). Even if it
was expected, it can be considered as an interesting result because it shows the consistency between the POLDER AOT retrievals and the ERA-Interim meteorological parameters. The wind speed is generally 1-2 m.s$^{-1}$ higher in case of larger absorption AOT and the $q_v$ is 4 to 6 times larger for these cases.

To conclude, few meteorological differences are found between the *high* and *low* situations at lower altitudes and where the clouds reside: similar temperature inversion, similar wind direction and specific humidity. However, humidity close
to the surface is higher for the *high* cases. At 700 hPa the wind direction and moisture are very different for the two situations: easterlies are associated to larger AOTs and larger humidity values, while the wind coming from the open ocean is characterized by low values of AOT and humidity.

## 4    Results

### 4.1    Difference in cloud properties

Following our strategy to make a distinction between *low* and *high* aerosol loads over the sample area, for June-August 2008, we will use the observations to analyse the variation of cloud parameters to different scenarios of aerosol and water vapour loadings, having in mind that the above-cloud aerosol load and the meteorological parameters might affect cloud properties. The variation of cloud parameters is analysed as a function of longitude and as a function of surface meteorological parameters.

Figure 9a and c present different cloud properties as a function of longitude for the two AOT conditions. We observe an evolution of some parameters that vary fundamentally in the same way with longitude, regardless of the *high* or *low* situations, which may be linked to the air mass transport. The MODIS $r_{eff}$ and the LWP increase westwards (panels a and c).





This might be a result of the evolution of the clouds optical and microphysical properties, as they are driven further away from the coast by the air masses (it is consistent with the fact that the winds are mostly westwards).

Furthermore, most of the parameters display a difference between the *low* and *high* situations, as there is a visible gap between the values corresponding to the two situations, independently on longitude. MODIS $r_{eff}$ increases from E to W from 6.5 to 11 μm when the aerosol loading is low. One can also observe an increase when the aerosol loading is high, but the increase is weaker, from 8 to 10 μm (Figure 9a). This difference may suggest that the cloud microphysics is different for the situations with and without aerosols. We notice thicker clouds when the absorption AOT is larger than 0.04, as POLDER COT increases by approximately 3 along the longitude (not shown here) and with LWP systematically larger by approximately 20 g.m$^{-2}$ (Figure 9c). Another cloud parameter we analysed is the POLDER cloud altitude $ZO_2$ (Figure 9b). In case of *low* AOT situations, $ZO_2$ slightly increases with longitude what may be the result of enhanced convection away from the coast. In the case of *high* AOT situations, we observe slightly lower cloud altitudes that remain under 1 km. The atmospheric subsidence shown in Figure 7d cannot explain the difference in altitude, as the cloud altitudes retrieved from POLDER measurements are lower for the polluted situations, for which the subsidence is lower. Therefore, it is unlikely that the lower cloud altitudes are a result of stronger atmospheric subsidence. The difference in altitude is weak, and the effect of an upper level of aerosol on the cloud altitude derived from oxygen pressure is questionable.

The specific humidity also presents different behaviours as a function of longitude between the polluted and less polluted situations (Figure 9d). We notice that when large loads of absorbing aerosols are present above clouds, the humidity at 925 hPa is rather constant along the longitude with values between 8.5 and 10 g/kg, while in the *low* case the humidity increases westwards from 5.5 g/kg to 9 g/kg.

We also analysed the LWP as a function of the sea surface temperature (SST) and the surface wind speed. As in Figure 9c, the results show a difference in the liquid water path for the two scenarios regardless of the meteorological conditions, with LWP being systematically larger for the *high* cases. LWP is increasing with surface wind speed (Figure 10a) but behaving very differently as a function of SST (Figure 10b). For the *low* case, LWP is clearly decreasing with SST, while a tendency is much less clear in the *high* case. The analysis of the cloud parameters variation over the zone shows a common feature of their variations (a westward increase of cloud parameters), and significant differences between the *low* and *high* aerosol situations with a higher LWP for the *high* case. This difference remains for different meteorological conditions (wind, SST). These results do not contradict a "cloud thickening" effect described by Wilcox (2010) who obtained from microwave measurements an increase of LWP by 20 g.m$^{-2}$ and around 200 m decrease of the cloud top altitude retrieved with CALIOP. Still, the result of Figure 9d suggests as a first analysis that the higher level of humidity in the *high* case could explain the higher amount of LWP.

The last covariance that we studied is the one between specific humidity at 925 hPa and LWP. The correlation should exist and be positive, based on the physical ground that more water vapour in the convective lower layers should be generally associated with more condensed water and LWP. Figure 10c shows the LWP retrieved for the *high* and *low* situations, as a function of the specific humidity. We notice that in both cases, the LWP increases linearly with $q_v$ up to 9 g.kg$^{-1}$, where a shift



occurs in the *low* case and the LWP is decreasing, while in the *high* case the LWP continues to increase. The relationship between LWP and $q_v$ for the *low* case seems well illustrating the strong cloud–radiation–turbulent–entrainment feedback (Zhu et al., 2005) that would exist for stratocumulus cloud field. Wood, (2012) describes this negative feedback as the following: a thickening cloud drives stronger entrainment [of warm, dry air], which results in cloud thinning. The interesting result that the behaviour is different in the *high* smoke case, with a continuous increase of LWP with $q_v$, suggests that processes leading to the negative feedback could be significantly perturbed by the presence of the overlying aerosol layers and the accompanying increased water vapour in above-cloud layers. In the next section, we perform radiative transfer calculations in order to quantify the difference in radiative heating rates between the *high* and *low* cases.

## 4.2    Overall radiative impact of the smoke layers.

Our hypothesis is that a smoke layer elevated above the clouds can induce a semi-direct effect through a change in the radiative budget above the cloud and/or at cloud top. More precisely, the presence of a layer-containing aerosol within a moisture profile will modify the radiative heating rates in the atmospheric column, as a consequence of radiative processes that are absorption, scattering and emission of radiation. It is also the case when high-level clouds overly stratocumulus clouds (Christensen et al., 2013). If the heating rates vary at the cloud level, they will directly impact cloud processes and development. Stratocumulus clouds are characterized by a strong cloud top cooling, which is the result of a radiative and evaporative cooling. Both act as drivers to create cloud top turbulence, mixing in the cloud layer, and small-scale dry air entrainment (Zuidema et al., 2009). Modification of radiative heating rates at the cloud top and its surrounding would eventually modify cloud top processes, the thermodynamic state of entrained air and its impact on clouds (Bretherton et al., 2004).

Figure 11 presents our results of the smoke layer's overall radiative effect estimated over the sample area close to Angola's coast and for the entire period of June to August 2008. The simulations were performed for individual CALIOP tracks, thus considering the variability of aerosol and cloud properties as well as their different altitudes.

The shortwave radiative transfer results under the CALIOP track show that over the sample area, the instantaneous shortwave direct radiative effect (DRE) at the top of the atmosphere (TOA) has values between 50 and 120 Wm$^{-2}$, with an average of 66.7 ± 23.2 Wm$^{-2}$. The maximum value corresponds to the largest POLDER AOT$_{865nm}$ and COT (not presented here). These positive values of DRE show that the aerosols reduce the local albedo by absorbing solar radiation, generating a radiative warming of the atmospheric column.

Figure 11 presents the mean profile of the shortwave, longwave and total heating rates (K/day), for *unpolluted* situations (in which we consider only the water vapour profiles, panel a) and for smoke events (panel b). This gives a better appreciation of the average aerosol and water vapour contributions on the radiative budget in the area. We computed the heating rate initially for individual profiles, and then we averaged over the entire region. This explains the different peaks between 0.5 and 1.5 km, as the cloud top altitude varies for each profile.

We observe that most of the warming in the atmosphere occurs where the smoke resides (between 2.5 and 4.5 km). The HR$_{SW}$ maximum value is 9 K/day, of which the water vapour contributes with 3 K/day (seen in panel a). The heating





observed above 4.5 km is due to the shortwave radiation absorbed by the water vapour; at the cloud level the warming comes from solar absorption by water vapour and cloud droplets. There is a longwave cooling at the cloud level, with mean $HR_{LW}$ of approximately -18 K/day and in the upper atmosphere a mean cooling up to -5 K/day. In case of polluted situations, the budget of the heating rates at the aerosol level shows that the aerosols warm the layer with an average $HR_{total}$ of 6 K/day, while water

vapour has an overall effect close to null over the cloud layer, with a compensation between its induced solar heating and infrared cooling. At the cloud level the mean $HR_{total}$ has negative values: in the *unpolluted* cases the $HR_{total}$ at cloud top reaches -13.5 K/day, while in the presence of above cloud aerosols, the value increases to -12.9 K/day. This difference represents the average effect of aerosol above cloud on the cloud top heating rate, as they absorb part of the shortwave radiation. At surface the longwave absorption leads to a strong warming, with values larger than 20 K/day, due to the absorption of the ocean's

longwave contribution by the cloud and water vapour.

Overall, we observe that the sample area is globally under the energetic influence of absorbing aerosols, leading to warming at the altitudes where aerosols reside. They create also a global positive shortwave direct effect at the top of the atmosphere. Finally, we observe that over the sample area there is a radiative cooling at the cloud top layers, slightly different between the *unpolluted* case and the case with aerosols.

However, in order to understand the aerosol and water vapour effects we have to account for the covariance between humidity and aerosol instead of only simulating cases with and without aerosols in the atmospheric column.

### 4.3    Distinction between water vapour and aerosol effects

The radiative study over the sampled area has already given us information on the average aerosol and water vapour contribution to the heating rates, mainly at the aerosol level. In this section, we analyse two different simulations for average

*high* and *low* loadings of aerosol /water vapour. We define a cloud type characterized by $r_{eff}$ of 10 μm and COT of 10, in order to consider the same contribution of the cloud droplets multiple scattering and heating rates in both situations. The objective is to investigate how different aerosol loadings and meteorological parameters could influence the heating rates and to separate in the end their different contributions and effects.

Figure 12 presents the first simulation, for which we calculated the heating rates in shortwave and longwave spectrums

and their total by considering the average aerosol properties in case of *high* and *low* situations (see Table 1) and the associated water vapour profiles (see Figure 7c). The left panel shows the heating rate profiles for polluted situations - large AOT and water vapour content; the middle panel presents the heating rate of the less polluted situations - low AOT and water vapour, and the right panel presents the difference between the resulted heating rates of the *high* and *low* cases. We observe a strong cooling at the cloud top level (at 1 km), with values that reach -70 K/day (panels a and b) and a warming at the smoke level

(between 2 and 4.5 km), more visible for the polluted case. The net heating rate profile (panel c) shows a maximum warming of about 6 K/day and a warming at the cloud top with 5 K/day between the *high* and *low* scenarios. These effects can be attributed to the presence of a smoke layer above the cloud, as the cumulative effect of aerosols and water vapour at the smoke level and at the cloud top.





In order to separate the water vapour from the aerosol contributions, we performed a second simulation for which the above-cloud AOT is the same (0.21 at 865 nm), but with distinct typical water vapour profiles for high and low cases. Therefore, the aerosol radiative contribution is equal for the two cases, and the subtraction of the heating rates obtained for the two profiles will provide the water vapour only radiative contribution in the column. Results are shown in Figure 13. In the shortwave, the difference of HR computation shows that the increase of water vapour has a warming effect in the aerosol layer of only 0.8 K/day, and a cooling effect in the cloud layer, due to a shadowing effect. At the cloud top the water vapour is the main agent to drive the net warming of about 5 K/day, which is equivalent to 7 % loss in cloud top cooling. The synthesis between the shortwave heating rates computed in the first simulation (water vapour and aerosol contribution) and those computed in the second scenario (water vapour only after the subtraction), provides the distinct contributions of moisture and aerosols in the heating rate difference between polluted and less polluted situations (see Table 2 and 3).

## 4.4    Discussion

These radiative transfer simulations help us understand the water vapour radiative contribution in the shortwave and longwave spectrum, and to separate its effects from those of the aerosol loading in the visible domain. It demonstrates that the aerosols accompanied by water vapour impact not only the layer where they reside, by heating or cooling their environment, but also have a distant effect on the underlying cloud systems. Results show that overlaying aerosols drive the warming within the aerosol layer, while the water vapour is responsible for a significant reduction of the cloud top radiative cooling. These effects eventually affect the cloud evolution in a way that could lead to a cloud thickening.

At this point we can attempt to explain the differences in the cloud properties observed in Section 4.1. If we consider again Figure 10c, we notice that for the same value of specific humidity at 925 hPa we have different behaviour of LWP between polluted and less polluted situations. These differences could result from a perturbation of cloud top processes (turbulence, entrainment) and from differences in the thermodynamics of above cloud layers. In the nonpolluted case, and as the cloud thickens, mixing and turbulence increase. The stronger entrainment that follows would result in a thinning cloud (Wood, 2012). We can notice this tendency in Figure 10c for the *low* cases, where the values of LWP increase up to 55 g.m$^{-2}$ followed by a decrease.  In the polluted case instead, where larger AOTs and water vapour are present above the cloud, the cloud top radiative cooling is reduced (mainly by water vapour radiative effects), the layers above the clouds are warmer (mainly due to SW absorption of aerosols), that reinforces the temperature inversion (see Figure 9b and Figure 12c). Thus, a combined effect of increased atmospheric stability and of cloud top cooling reduction would eventually lead to a decrease of entrainment rate and its moderating effect on clouds, all the more because the air that is entrained would be moister. It would explain a cancellation or reduction of the cloud thinning effect, illustrated in Figure 10c with a linear increase of LWP with moisture for humidity higher than 8 g.kg$^{-1}$. This study indicates the importance of separating the aerosol and water vapour contributions when studying cloud adjustments to above cloud aerosols.



## 5    Conclusions and perspectives

In our study, we focused on the impact of absorbing aerosol layers overlying a lower cloud layer, their associated radiative forcing and their potential effects on the underlying clouds. The main objective was to disentangle the effect of aerosols on clouds from the meteorological effect, and to calculate their radiative impact. The region of interest was the South Atlantic Ocean due to the large loads of African biomass-burning absorbing aerosols that are frequently transported over the main South Atlantic stratocumulus deck.

We realized a synergy between CALIOP and POLDER measurements of aerosols above clouds, to which we added meteorological parameters provided by the ERA-Interim reanalysis. We analysed the properties of aerosols and clouds for the period May - October 2006 to 2009, in particular the information about the vertical distance between the two layers: along the longitude, the clouds develop vertically as they are transported further westwards; the aerosols are mainly detached from the cloud close to the coast and afterwards they lose altitude, which is explained by the wet and dry deposition processes and the atmospheric circulation over the South of Africa and Southeast Ocean. The evolution of aerosol properties travelling westward (increase of their size, decrease of their absorption and Ångström exponent) suggests an aging of aerosols.

For a more detailed analysis we selected a small area close to the coast of Angola that is near the main stratocumulus deck identified in the region, and where large loads of aerosols are transported at higher altitudes, mainly detached from the clouds below. The study was limited to three months (June to August 2008) in order to constrain the meteorological variability that can affect the clouds and the overlaid aerosols, but also to maintain a sufficiently large dataset for a better statistical analysis.

A first interesting result of our approach is that we are able, using a CALIOP / POLDER / ERA-Interim synergy and a radiative transfer model, to provide advanced estimates of the aerosol radiative forcing and the vertical heating rate profiles when aerosol layers are above lower clouds, in both the shortwave and longwave domains. Over the area, we estimated that smoke layers located above clouds significantly perturb TOA net flux and the solar illumination of clouds at the time of the overpass of POLDER (13:30 UTC). The study revealed positive average values of the DRE at TOA, between 50 and 120 Wm$^{-2}$, with an average of 66.7 Wm$^{-2}$ and a standard deviation of 23.2 Wm$^{-2}$, which signifies a reduced scene's albedo and a warming of the atmospheric column strongly correlated to the aerosol loading. The shortwave heating rate shows a warming of 9 K/day at the aerosol layer, of which 3 K/day are due to the water vapour shortwave absorption. However, the total effect of water vapour is almost null, as the shortwave heating is compensated by its longwave radiative cooling. Another estimate is the heating rate at the cloud top layers. We observed a net cloud top radiative cooling dominated by infrared transfer, around -60 K/day for each pixel under the CALIOP track, and a mean of -13 K/day over the area, as clouds are diversely vertically located. Removing aerosols from the calculation show a slight increased cooling. In the total balance between shortwave and longwave heating rates, we notice a small effect of the presence of aerosols above clouds (of 0.56 K/day), which is explained by the shadowing effect of aerosols, that attenuate the solar radiation to reach the cloud top.



For this area and for this time period, we observed a strong covariance between the increase of the specific humidity and the increase of AOT, especially at 850 and 700 hPa (Figure 6). One explanation could come from the release of water vapor together with aerosols from the African sources (Parmar et al., 2008). Another explanation, not explored here, could come from the rapid adjustments of water vapour to the presence of aerosols (Smith et al., 2018).

This covariance motivated us to analyse the correlation between cloud and meteorological parameters for two distinct regimes: highly absorbing aerosols and aerosol layers with low absorption. This distinction confirmed two different humidity profiles characteristic of *high* and *low* situations. At 700 hPa, the difference clearly originates from two distinct air masses: the high loads of aerosols come from the land (direction east-west), with high amounts of water vapour, while the low cases originate from the open ocean, with lower moisture quantities. The analysis of meteorological conditions at 925 hPa shows

some differences, but mainly common features of wind directions, temperature inversion and humidity. The larger difference was observed above the cloud layers, at 700 hPa, where the aerosol layer resides.

Our results confirm previous satellite observations and studies that showed that clouds contain more water and are at slightly lower altitudes when large loads of absorbing aerosols are located above them. Indeed, we observed a significant increase of LWP between *low* and *high* cases, whatever the meteorological conditions (Figure 9 and Figure 10).

The cause and effect of the cloud "thickening" is a challenge to untangle. Stabilization of above-cloud layers that experience a warming – which should lead to temperature inversion reinforcement, decrease of air entrainment and humidity preservation – have been suggested in previous studies. Our results of layer warming due to aerosol SW absorption is not contradictory, but aerosol layers seem here quite distant from the cloud top to confidently confirm this explanation. For the two smoke scenarios, two distinct relationships have been obtained between the water vapor at 925 hPa and the LWP: a

common high and positive correlation is observed for LWP below 55g.m-2 in both cases but, above this level, the correlation is continued only for the high case. This suggests that the negative feedback evoked by Wood (2012) (a cloud thinning that follows a stronger entrainment, consequence of a cloud thickening) is attenuated, if not cancelled, in the high case. A possible explanation that we promote here is a reduction of cloud top entrainment and its consequence due to two factors: a decrease of cloud top turbulence due to a decrease of cloud top cooling, and that air entrained is more humid in the high case.

We evaluated the difference in cloud top cooling between the *low* and *high* scenarios by performing two radiative transfer simulations with two typical covariant loadings of aerosol and humidity. The results of the radiative transfer simulations show that, while an increase in aerosols is 90 % responsible for increasing the heating of the layer where they reside, the increase of the water vapour amount is responsible for a decrease of cloud top cooling of + 4.7 K/day (7%), while a slight increase of the aerosol's shadowing leads to a cooling of -0.23 K/day. Thus, the presence of smoke layers in the scene significantly impacts

its radiative budget and modifies the heating rates, with an increase of heating in the aerosol layers, and a decrease of the cooling at the cloud top, which may significantly impact the cloud top dynamics. Accounting for the covariance between aerosol loading and water vapour seems important for understanding the observed cloud thickening (these are identified mechanisms for AAC semi-direct effects on clouds).




As perspectives, we propose to validate our computed heating rates by comparing them with the AEROCLO-sA, CLARIFY and ORACLES airborne data. During these field campaigns, various lidar and polarimeter were deployed and could be used to estimate in synergy the heating rates profiles. An exercise of validation would consist in validate the heating rates with the concomitant flux measurements performed at different level through the smoke layers. Furthermore, we plan to

combine our database with a climate or regional model to check the consistency of our data. The model can be constrained using observations (aerosol and cloud parameters, water vapour content, meteorological profiles), and can provide simulations made with and without aerosols and to help disentangle the effects of meteorology versus the aerosol effects. Such new strategies combining models and active/passive remote sensing data with meteorological data and airborne sounding will help to better understand the impacts of aerosols on the clouds and climate at regional and global scale. Using our large database,

we can further calculate the heating rates of aerosols above clouds for a larger region to better study the transport effect, or even at a global scale.

Author Contributions.

LD, NF and FW conceived the study. FW and PG acquisitioned the funding. LD generated the database and analysed the data with the help of NF, FW and FP. NF analysed the meteorological data and helped interpreting the changes in cloud properties. FW modified the GAME models in VIS and IRT for the purpose of this study and defined the "high" and "low" strategy for investigating the aerosol effects on clouds. FT provided IT support. LD wrote the paper with reviews from all authors.

Data availability.

All data are available at ICARE website on-line archive: http://www.icare.univ-lille1.fr/archive. POLDER aerosol above cloud products and CALIOP extinction profiles can be obtained by email request to ICARE. The reanalysis data are distributed by ECMWF Meteorological Archival and Retrieval System (MARS).

Acknowledgments.

Lucia Deaconu's grant was provided by the CaPPA project (Chemical and Physical Properties of the Atmosphere), which is funded by the French National Research Agency (ANR) through the PIA (Programme d'Investissement d'Avenir) under contract ANR-11-LABX-0005-01 and by the Regional Council Hauts-de-France and the European Funds for Regional Economic Development (FEDER). The authors would like to acknowledge Philippe Dubuisson, for his insights on the GAME

model, Fanny Minvielle for helping to provide ECMWF data, and ICARE Data Service Centre for providing the satellite retrievals.



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



**Table 1: Mean values of aerosol properties over the specific area from June to August 2008, in case of HIGH values of absorption AOT (> 0.04) and in case of LOW values of absorption AOT (< 0.01), respectively.**

| POLDER aerosol properties | absorption $AOT_{865nm}$ | $AOT_{865nm}$ | $SSA_{865nm}$ | $AE_{670/865}$ |
|---|---|---|---|---|
| HIGH | 0.05 | 0.26 | 0.80 | 2.03 |
| LOW | 0.005 | 0.06 | 0.86 | 1.75 |

5        **Table 2: Shortwave and longwave heating rate (K/day) values for large and low AOT and specific humidity ($q_v$) values (first simulation) and for a constant AOT and high and low values of the specific humidity profile (second simulation), obtained at the cloud top level and at 4 km at noon.**

| | | HIGH or LOW AOT and $q_v$ [K/day] ($1^{st}$) | | | HIGH or LOW $q_v$, constant AOT [K/day] ($2^{nd}$) | | |
|---|---|---|---|---|---|---|---|
| | | $HR_{SW}$ | $HR_{LW}$ | $HR_{SW} + HR_{LW}$ | $HR_{SW}$ | $HR_{LW}$ | $HR_{SW} + HR_{LW}$ |
| Cloud top (1 km) | HIGH | 6.09 | -67.53 | -61.44 | 6.31 | -67.53 | -61.22 |
| | LOW | 7.88 | -73.79 | -65.91 | 7.69 | -73.79 | -66.1 |
| | HIGH-LOW | **-1.79** | 6.26 | 4.47 | **-1.56** | 6.26 | 4.7 |
| Aerosol layer (4 km) | HIGH | 9.86 | -2.86 | 7 | 8.78 | -2.86 | 5.92 |
| | LOW | 3.41 | -2.72 | 0.69 | 8.02 | -2.72 | 5.3 |
| | HIGH-LOW | **6.45** | -0.14 | 6.31 | **0.76** | -0.14 | 0.62 |

10        **Table 3: Subtraction of aerosol (A) and water vapour (WV) contributions in shortwave (SW) and longwave (LW) between the first and second simulation.**

| | Simulations subtraction: $1^{st} - 2^{nd}$ | |
|---|---|---|
| | In SW | In LW |
| Cloud top | A:    -1.79 - (-1.56) = -0.23 K/day<br>WV: -1.56 K/day | WV: 6.26 K/day |
| Aerosol layer | A:    6.45 - 0.76 = 5.69 K/day<br>WV: 0.76 K/day | WV: -0.14 K/day |





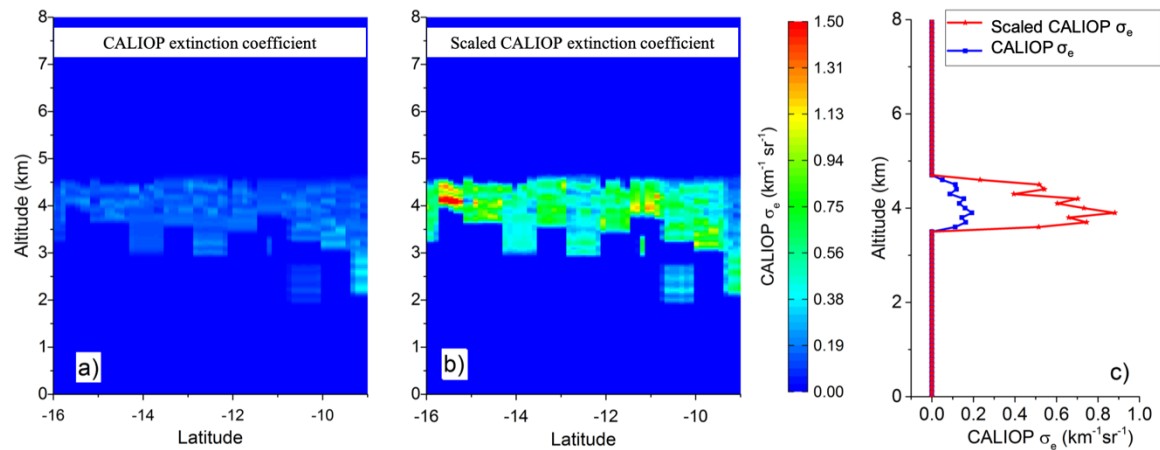

**Figure 1: Biomass-burning study case: (a) CALIOP initial extinction coefficient (m$^{-1}$) at 532 nm ($\sigma_{e,532nm}$), (b) scaled CALIOP $\sigma_{e,532nm}$ as a function of latitude; (c) example of CALIOP extinction coefficient profile at 532 nm (blue) and the result of scaling (red) with the POLDER AOT extrapolated at 532 nm.**







**Figure 2: Mean values of aerosol above cloud and cloud properties above the South Atlantic Ocean, for a period of six months (May to October) from 2006 to 2009: a) POLDER AOT$_{865\,nm}$. The black box represents the sample area close to Angola's coast described in Sect. 3.2; b) POLDER AE$_{670/865\,nm}$; c) POLDER absorption AOT$_{865\,nm}$; d) POLDER SSA$_{865\,nm}$; e) POLDER COT (corrected for above cloud absorbing aerosols); f) MODIS effective radius, r$_{eff}$ (µm); g) POLDER Cloud Altitude (ZO$_2$) derived from Oxygen Pressure (km).**

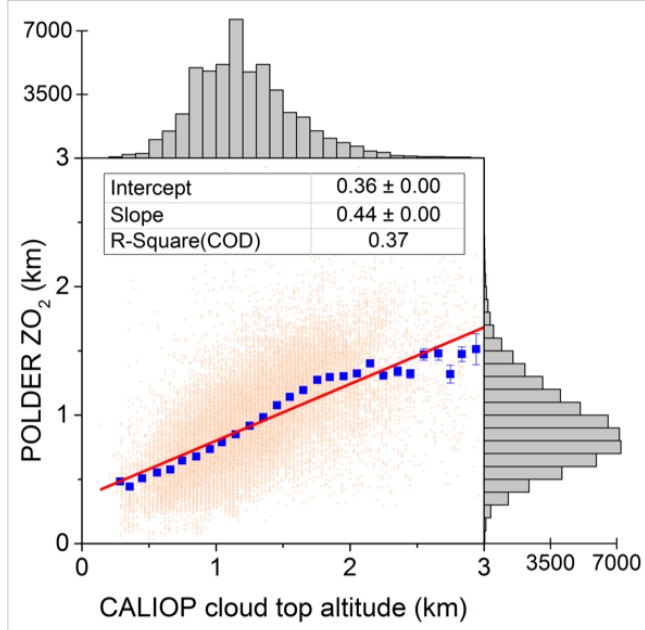

**Figure 3: POLDER cloud altitude (ZO₂) compared to CALIOP cloud top altitude, over a period of six months (May to October) from 2006 to 2010 along the CALIOP track. Lateral histograms show the data distribution.**



**Figure 4: First and second row of the panel present the histograms of the cloud top altitude (CTA), the aerosol top altitude (ATA) and the aerosol base altitude (ABA) as a function of longitude for two time periods: May to July and August to October, respectively, from 2006 to 2010. The mean, median and standard deviation over four degrees of longitude as well as the number of measurements are shown in the associated tables. The selected area extends from 30° S to 5° N and 12° W to 14° E over the South Atlantic Ocean (SAO).**





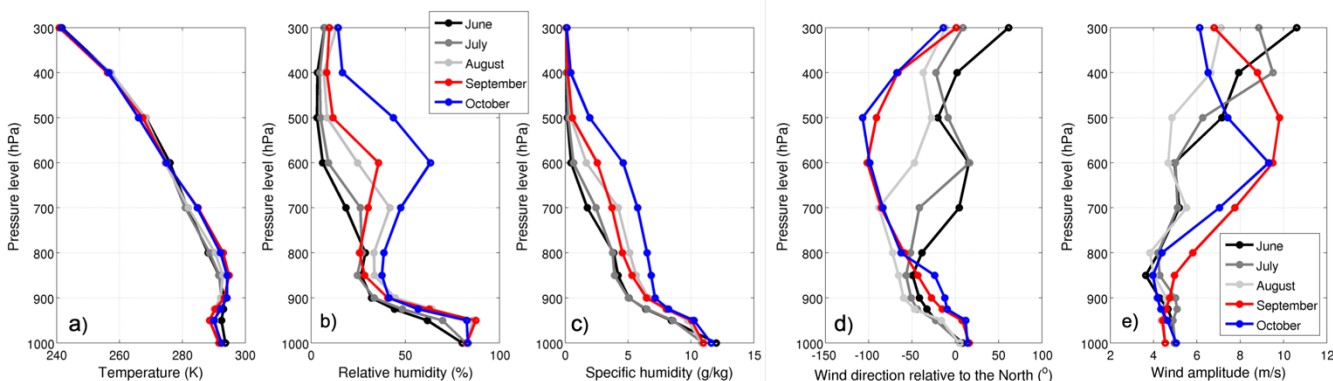

**Figure 5: Monthly mean meteorological parameters computed with ERA-Interim models at 12h UTC, from June to October 2008, over the sample area: a) temperature (K); b) relative humidity (%); c) specific humidity (g.kg$^{-1}$); d) wind direction relative to the North ($^O$); e) wind amplitude (m.s$^{-1}$).**

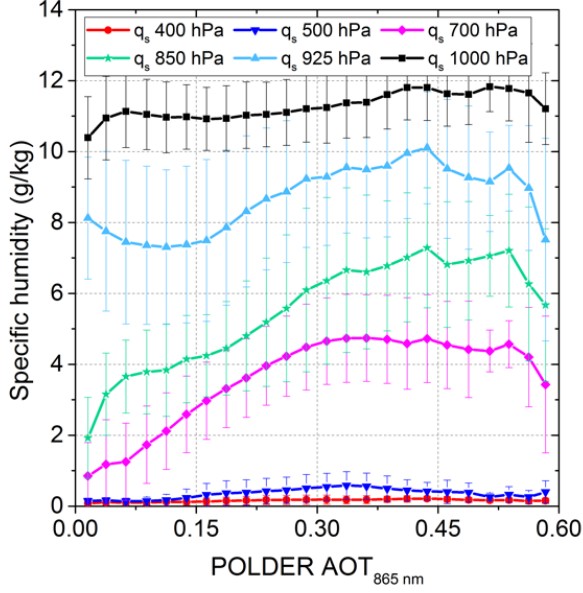

**Figure 6: Mean specific humidity as a function of POLDER AOT$_{865nm}$ retrieved at different pressure levels, within the selected region over June-August 2008. The vertical bars represent the standard deviation for the specific humidity.**





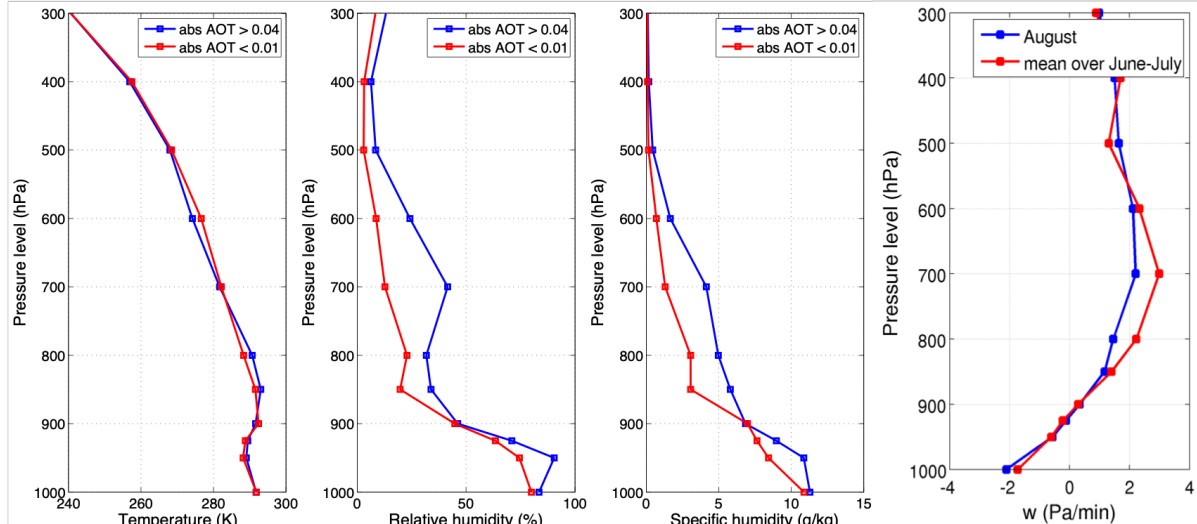

**Figure 7: Mean values of (a) temperature profile and (b) relative humidity profile, (c) specific humidity profile and (d) atmospheric subsidence at noon, w (downwelling wind, Pa/min), within the sample region, from June to August 2008 selected for two situations: POLDER absorption AOT$_{865nm}$ smaller than 0.01 (red lines, mainly June-July 2008) and POLDER absorption AOT$_{865nm}$ larger than 0.04 (blue lines, mainly August 2008).**

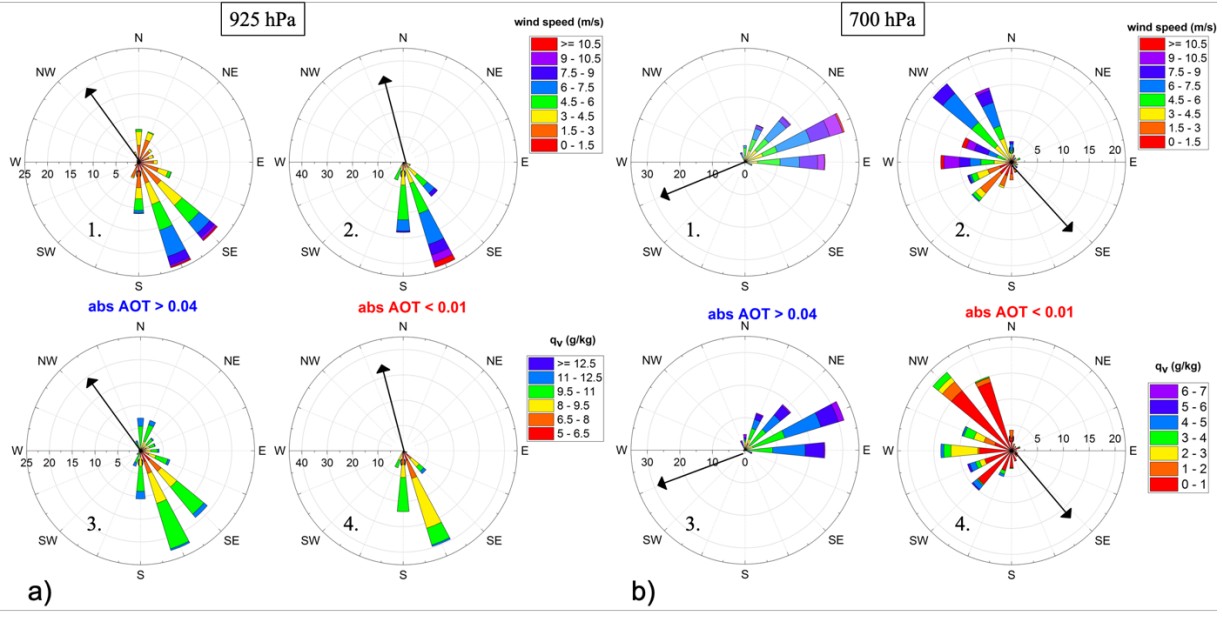

**Figure 8: Wind speed rose (1-2) and specific humidity rose (3-4) at 925 hPa (a) and 700 hPa (b) for situations with absorption AOT larger than 0.04 (1-3) and with absorption AOT smaller than 0.01 (2-4). The radius shows the frequency of wind direction. The arrow represents the main wind direction.**





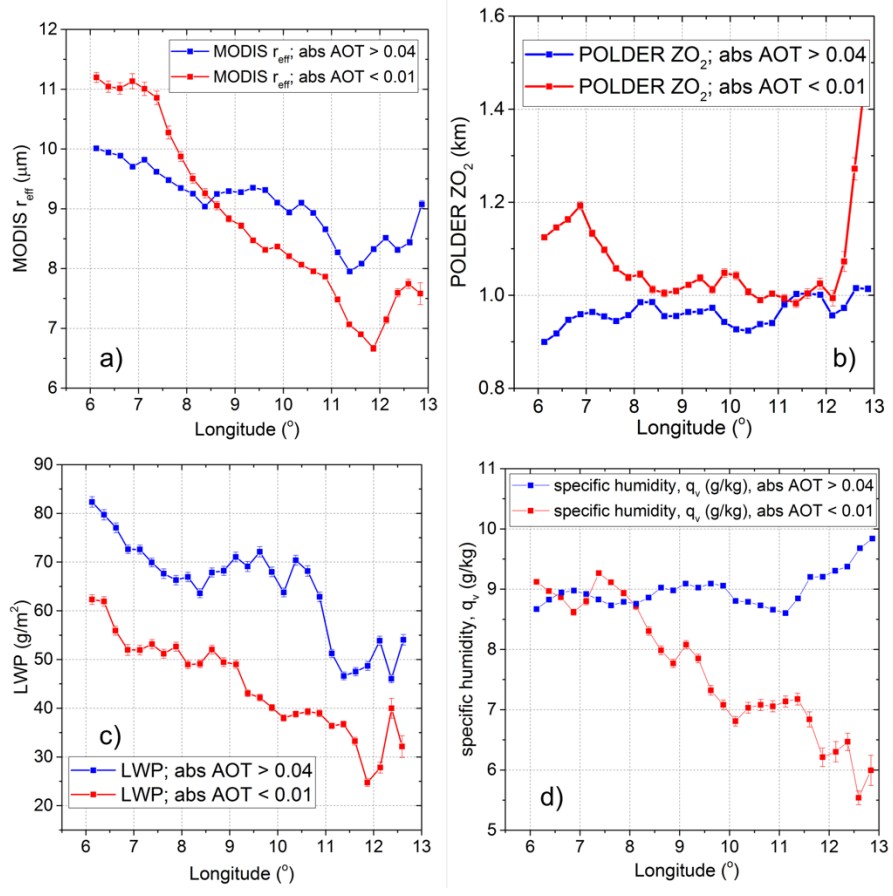

**Figure 9: Mean values of MODIS $r_{eff}$ (a) POLDER ZO$_2$ (b), Liquid Water Path (LWP) (c) and specific humidity at 925 hPa as a function of longitude. The data are separated into situations with POLDER absorption AOT$_{865\ nm}$ smaller than 0.01 (red lines) and with POLDER absorption AOT$_{865\ nm}$ larger than 0.04 (blue lines).**

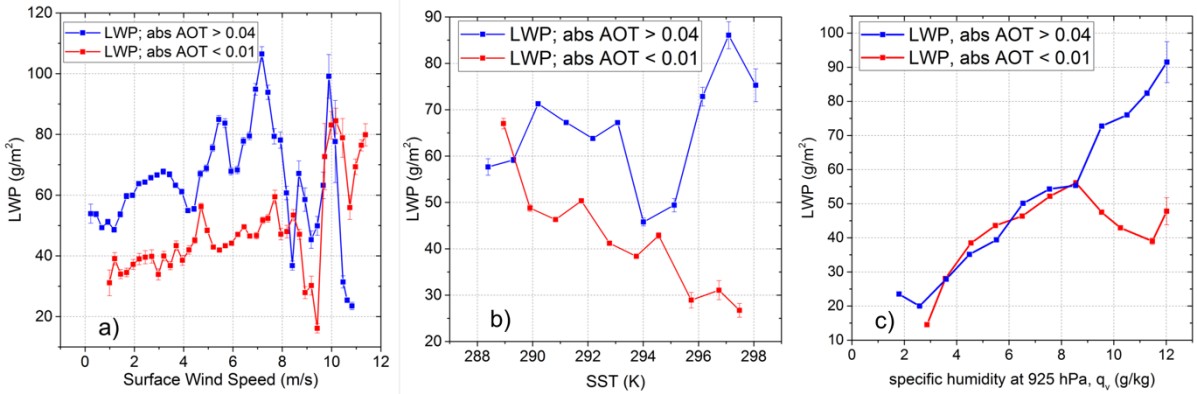

**Figure 10: Mean values of LWP as a function of meteorological parameters: surface wind speed (a), Sea Surface Temperature (SST) (b) and ERA-Interim specific humidity at 925 hPa. The data are separated into situations with POLDER absorption AOT$_{865\ nm}$ smaller than 0.01 (red lines) and with POLDER absorption AOT$_{865\ nm}$ larger than 0.04 (blue lines).**





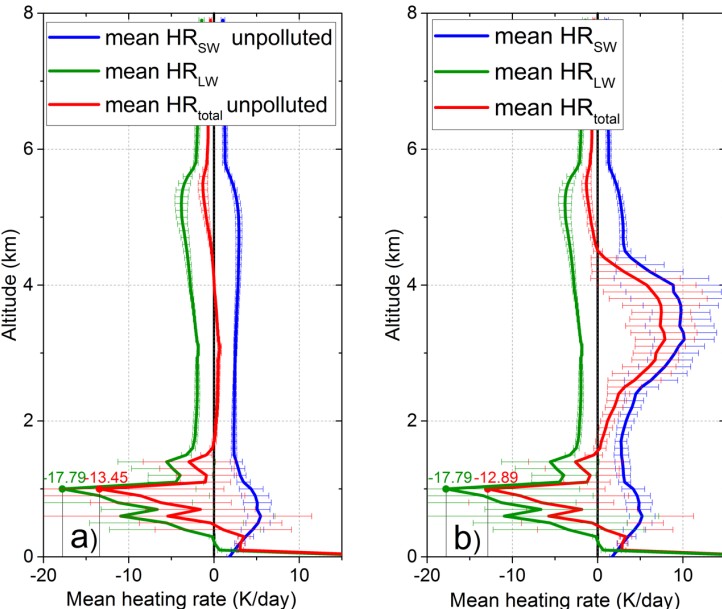

**Figure 11: Average heating rate for "unpolluted" cases (a) and polluted situations (b), over the sample area, from June to August 2008: shortwave HR (blue line), longwave HR (green line) and total HR (red line). The horizontal bars represent the standard deviation.**

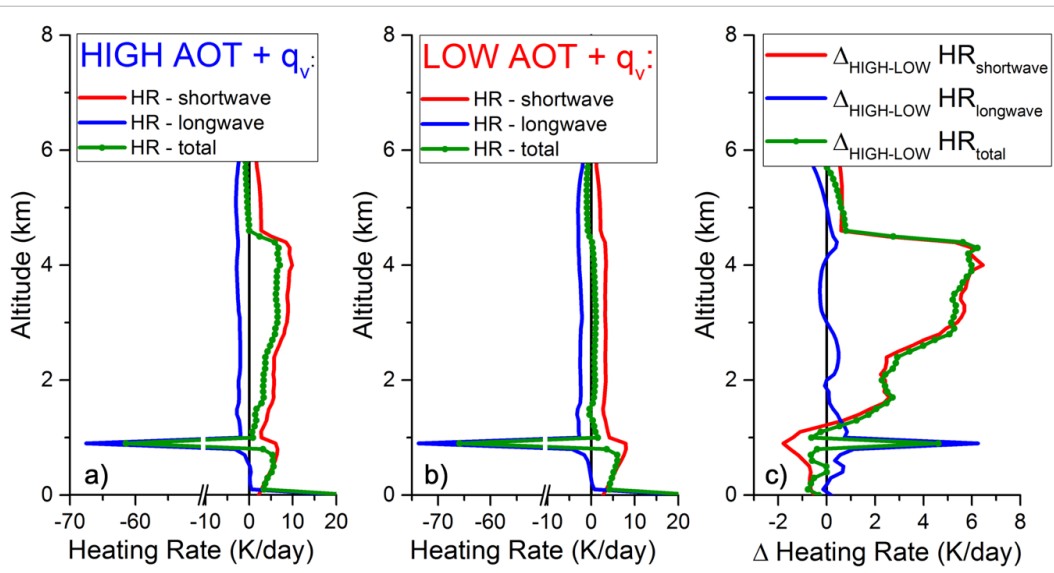

**Figure 12: Calculated heating rates profiles (K/day) in shortwave (red line), longwave (blue line) an total budget (green dot line) at 12 h, for (a) average HIGH cases ($AOT_{865nm} = 0.26$ and water vapour typical for HIGH situations), (b) average LOW cases ($AOT_{865nm} = 0.06$ and water vapour typical for LOW situations) and (c) the difference between the HIGH and LOW heating rate profiles.**





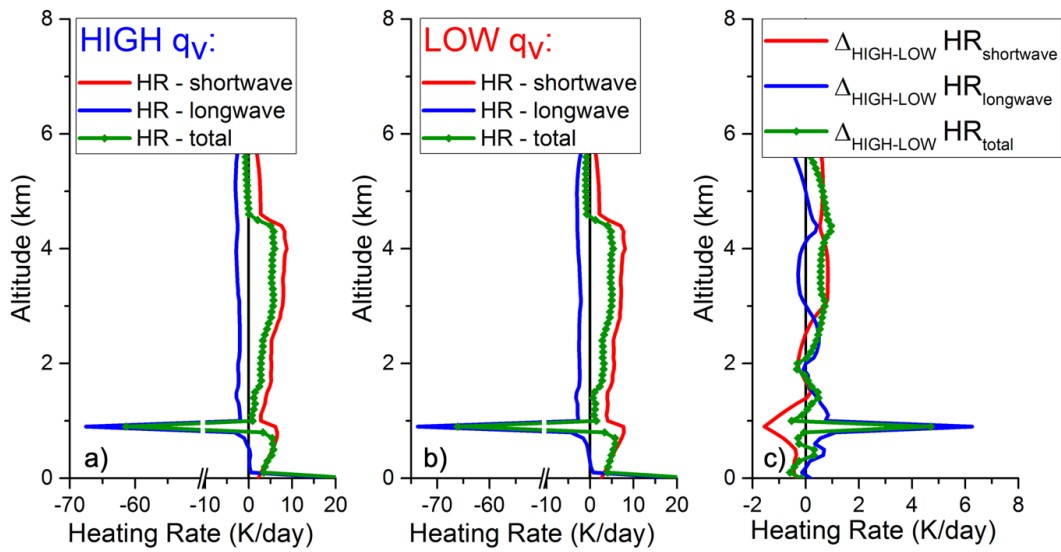

**Figure 13: Calculated heating rates profiles (K/day) in shortwave (red line), longwave (blue line) and total budget (green dot line) at 12 h, for an average AOT$_{865nm}$ over the region of 0.21 and (a) water vapour typical of the HIGH situation and (b) water vapour typical of the LOW situations, respectively. The last panel (c) presents the difference between the HIGH and LOW heating rate profiles.**