# Peer review of "Satellite inference of water vapor and aerosol-above-cloud combined effect on radiative budget and cloud top processes in the Southeast Atlantic Ocean"

_Atmospheric Chemistry and Physics, 2019_

## Referee Comment (RC1) · Anonymous Referee #1 · 23 Apr 2019

Review on "Satellite inference of water vapor and aerosol-above-cloud combined effect on radiative budget and cloud top processes in the Southeast Atlantic Ocean" By Deaconu et al.

This paper presents an analysis of the variation and covariation of aerosol and cloud properties, as well as meteorological conditions, in the SE Atlantic region based on satellite observations. Compared to many previous studies on this topic, this paper sheds an important light on the co-variation of water vapor and above-cloud aerosols, and the implications for radiative effects on cloud. Overall, this paper is well written

and well-organized. The topic is a good fit to ACP. On the other hand, it can be further improved if the following questions and issues can be addressed and clarified.

Comments/questions/suggestions: A few important papers on the above-cloud aerosols in the SE Atlantic region are missing from the Introduction. They should be cited.

o Page 2 Line 7~8: Swap et al. (1996) is perhaps the first one documenting the long-range transport of smoke aerosol from African continent to Atlantic.

o Page 2, line 16: Recently, Zhang et al. (2016) provide a nice analysis of how DRE of above-cloud aerosol depends on AOT and COT (see their Figure 9).

o Page 2, line 25: A few recent papers have studied the Twomey effect of above-cloud aerosols in the SE Atlantic region (Costantino and Bréon 2013; Lu et al. 2018). In particular, Lu et al. (2018), showed that the brightening effect due to entrained aerosols is actually stronger than the DRE and semi-direct effect.

o Page 4, line 5~15: A recent study by Zhou et al. (2017) provides a more comprehensive picture of the radiative interactions between above-cloud aerosols and low clouds in the SE region.

Section 2.1: The Single-scattering albedo (SSA) is used later for deriving the absorption AOT. Is SSA a retrieved parameter or an a-priori assumption in the retrieval?

POLDER retrievals are done at the 6x6 km. At this scale, a significant fraction of pixels would be partly cloudy. How are partly cloudy pixels treated in the analysis? If they are simply screened out, would that lead to sampling bias?

MODIS Reff retrieval is not really biased when above-cloud aerosol is present. But MODIS COT retrieval can be substantially biased (underestimated). See Meyer et al. (2013). It needs to be clarified how the COT retrieval bias is accounted for the LWP estimation.

Page 7, Line 25: Indeed, CALIOP retrieval of above-cloud smoke AOT can be significantly biased, as pointed out by the authors and many other previous studies. This is because CALIOP cannot "see" the low portion of a thick smoke layer, i.e., it puts the bottom of aerosol layer too high. Therefore, simply using the extinction profile for HR computation seems to be a bad idea. It needs to be clarified here why this is justified.

Page 8 Line 30: Again, is SSA a retrieved parameter or a pre-assumption. Is it a constant or it can vary spatially or temporally, and how?

Page 9 Line 10: It is interesting that the COT bias is pointed out here, but not in Section 2.1.

Page 13, The analysis in this section is somewhat superficial. Overall, it remains unclear after reading if those differences between low and high AOT loading cases in Figure 9 and 10 are significant and meaningful. Are they simply coincidence or there are some underlying physics? Significant revision is needed here. In particular, these differences should be put in the context of previous studies, e.g., Lu et al. 2018; Zhou et al. 2017

Page 14 Line 4~7: It remains puzzling to me why Figure 10c "seems well illustrating the cloud-radiation-entrainment feedback". It needs to be explained better and, in more detail, here. What is the underlying physics of this feedback? Why does the decrease of LWP with qv "illustrate" this feedback?

Page 15 Line 5: The total HR at cloud top for the unpolluted case is -13.5 K/day and the corresponding value for polluted cases is -12.9 K/day. Considering that the LW cooling rate for both groups is the same (i.e., -17.79 K/day). This difference indicates that the SW heating at the cloud top for the polluted case is actually stronger (i.e., more positive). Isn't this counterinitiative? Shouldn't the extinction of above-cloud aerosol layer reduce the SW heating rate at cloud top in the polluted cases? This SW heating difference should be pointed out here and explained in detail.

The overall values between the polluted and unpolluted cases in Figure 11 seem pretty close. Some statistical test (e.g., T-test) need to be performed to tell if the differences are statistically significant.   Costantino, L., and F. M. Bréon (2013), Aerosol indirect effect on warm clouds over South-East Atlantic, from co-located MODIS and CALIPSO observations, Atmospheric Chemistry and Physics, 13(1), 69–88, doi:10.5194/acp-13-69-2013.

Lu, Z., X. Liu, Z. Zhang, C. Zhao, K. Meyer, C. Rajapakshe, C. Wu, Z. Yang, and J. E. Penner (2018), Biomass smoke from southern Africa can significantly enhance the brightness of stratocumulus over the southeastern Atlantic Ocean, PNAS, 115(12), 201713703–2929, doi:10.1073/pnas.1713703115.

Meyer, K., S. Platnick, L. Oreopoulos, and D. Lee (2013), Estimating the direct radiative effect of absorbing aerosols overlying marine boundary layer clouds in the southeast Atlantic using MODIS and CALIOP, Journal of Geophysical Research-Atmospheres, 118(10), 4801–4815, doi:10.1002/jgrd.50449.

Swap, R., M. Garstang, S. A. Macko, P. D. Tyson, W. Maenhaut, P. Artaxo, P. Kållberg, and R. Talbot (1996), The long‐range transport of southern African aerosols to the tropical South Atlantic, J. Geophys. Res., 101(D19), 23777–23791, doi:10.1029/95JD01049.

Zhang, Z., K. Meyer, H. Yu, S. Platnick, P. Colarco, Z. Liu, and L. Oreopoulos (2016), Shortwave direct radiative effects of above-cloud aerosols over global oceans derived from 8 years of CALIOP and MODIS observations, Atmospheric Chemistry and Physics, 16(5), 2877–2900, doi:10.5194/acp-16-2877-2016.

Zhou, X., A. S. Ackerman, A. M. Fridlind, R. Wood, and P. Kollias (2017), Impacts of solar-absorbing aerosol layers on the transition of stratocumulus to trade cumulus clouds, Atmospheric Chemistry and Physics, 17(20), 12725–12742, doi:10.5194/acp-17-12725-2017.

---

## Referee Comment (RC2) · Meloe Kacenelenbogen (Referee) · 2 May 2019

The authors use a combination of POLDER, MODIS, CALIOP and modeled meteorological profiles to evaluate the changes in met. parameters (e.g., Temperature, RH, Specific Humidity, Winds), cloud properties (droplet effective radius, top height, liquid water path), and heating rates as a function of more or less overlying AAOD. This paper is of good quality, well written and structured. It will be worthy of publication, once the issues below are addressed.

Overall comments: • Section 2.1. could benefit from a Table listing all the products and corresponding satellites/ models used in their method. • The authors base their study near the coast because this is where "aerosols are mainly detached from clouds" using CALIOP (by the way, CALIOP will likely miss the base of the aerosols). But then, further in their study, they analyze potential aerosol-cloud interactions. It would be worth adding some information on aerosol-cloud contact frequency over the region • The reader could benefit from an explanation of their AAOD thresholds (i.e., >0.01 and <0.04); If AAOD is the threshold, and it says "high" or "low" aerosol loading, does this mean that the authors assume a constant SSA value? If not, shouldn't they say "higher loading and higher absorption" instead?

Detailed comments: . I had to read the title multiple times to understand it. "Combined effects of water vapor and aerosols on underlying cloud top processes and radiative budget from satellites over the South East Atlantic Ocean" or something along those lines would make it clearer. . P1, line 14: "it is a prerequisite" . P1, line 21: "sensing techniques" . P2, Line 10: "negligible wet scavenging" needs more references . P3, line 11, line 30 (and other places): should read "cloud properties", "particle size", "droplet effective radius" etc.. . P3, line 20: I suggest briefly describing the "assumptions" (i.e., mostly the CALIOP lidar ratio) . P3, line 21: when introducing the depolarization method, I suggest saying "first introduced by Hu et al., 2007 and further implemented by e.g., Liu et al., 2015, Kacenelenbogen et al., 2019 Deaconu et al., 2017" . P3, L22: "AAC properties" . P3, L31: I suggest mentioning SSA from Peers et al.(2015) is retrieved above clouds. . P4, L3: please consider referring to Table 1 or 2 of Kacenelenbogen et al. [2019] . P4, L15: "high loadings of smoke" . P5, L14: I suggest describing the "semi-direct effect" . P5, L30: available at 490 and 865 nm . P5, L31: I suggest briefly describing the Angstrom exponent . P5, L32: "the aerosol model prescribed in the POLDER (?) satellite algorithm" . P6, L6: Are MODIS cloud properties corrected for AAC? . P6, L7: "cloud altitude derived from POLDER (ZO2)" . P6, L7: "ZO2 is calculated using. . ." . P6, Section 2.1: As said in the overall comments, this section could benefit from a Table listing all the products and corresponding satellites/

models used in the method. . P7, L3: "The GAME model" . P7, L4: Instead of "for this", I suggest "inputs to GAME are. . ." . P7, L10: Instead of "we", I suggest "GAME uses" . P7, L12: Instead of "the CALIOP method", I suggest "the standard CALIOP product can underestimate. . ." And this is happening also when aerosols are below a certain detection threshold. You could reference Kacenelenbogen et al. [2014]. . P7, L24: "using the POLDER method" . P7, L25: I suggest mentioning that, although this might not affect your study, the aerosol base height might still very likely be biased high after scaling the extinction profile as seen on Fig. 1 . P8, L7: Consider replacing easterlies by easterly winds . P8, L27: I suggest "decrease" instead of "go down" . P8, L28: I suggest "value prescribed for the dust model in the POLDER algorithm" if I understand this correctly. . P9, L1: For SSA values during ORACLES-2016, I suggest referencing Pistone et al., [2019] . P9, L13: I suggest describing Fig. 2g before Fig. 3 . P9, L17: "lower than" . P9, L19: "Nevertheless, the stratocumuli are low-level clouds, so, an underestimation of around 300 m by the POLDER product is more likely": this is not clear to me. . P9, L21: "stratocumulus become more fractioned" would it be worth showing the cloud fraction as well? . P10, L9: I suggest explaining the thresholds of 0.01 and 0.04 for AAOD. Could it be an AOD of 0.2 at 865nm with an SSA of 0.8? . P10, L17: "aerosols are mainly detached from low level clouds": I suggest to be more quantitative. Again, CALIOP aerosol base height is biased high. There is likely more contact than what is seen from CALIOP. Maybe use results from Rajapakshe et al. [2017]? . P10, L31: "June to August (JJA) 2008" . P11, L7: "from 7.5 to 10 g.kg-1" . P11, L8: "smoke plume level (i.e., between 850 and 700 hPa)" . P11, L18: "plumes resulting from" . P11, L30: "Figure 7b and 7c" . P12, L1: I suggest "smoke" instead of "polluted" (urban pollution and smoke being two different aerosol types when using satellite remote sensing) . P12, L25: the authors choose the sampling area so that "aerosols are mainly detached" from clouds. Are you implying more aerosol-cloud contact here? . P 13, L1: This is not clear. I would rephrase. . P13, L14: I would quantify how low the difference is. . P13, L20: "wind speed (see figure 10)" . P17, L12: "South East Atlantic Ocean" . P17, L13: "increase in size, decrease in absorption" . P17, L20: why

"advanced"?; replace "forcing" by "effects" . P17, L21: I would quantify "lower" . Fig. 4: I suggest adding that these results use CALIOP data in the legend; First row could say MJJ and second row could say ASO . Fig. 5, 6, 7: An illustration of the mean aerosol and cloud layer heights for the sampling domain and period would help the reader . Fig. 7d: Legend is confusing: is it now blue for august and red for June-July? . Fig. 9: It would not hurt to remind the reader that this is about clouds only and add "at 925hPa" to the y-axis of Fig. 9d . Fig. 7, 9, 10: I find it non-intuitive to color the "low" aerosol loading conditions in red and the "high" aerosol loading conditions in blue. I would have done the opposite

Deaconu, L. T., Waquet, F., Josset, D., Ferlay, N., Peers, F., Thieuleux, F., Ducos, F., Pascal, N., Tanré, D., Pelon, J., and Goloub, P.: Consistency of aerosols above clouds characterization from A-Train active and passive measurements, Atmos. Meas. Tech., 10, 3499–3523, https://doi.org/10.5194/amt-10-3499-2017, 2017. Kacenelen-bogen, M. S., Vaughan, M. A., Redemann, J., Young, S. A., Liu, Z., Hu, Y., Omar, A. H., LeBlanc, S., Shinozuka, Y., Livingston, J., Zhang, Q., and Powell, K. A.: Estimations of global shortwave direct aerosol radiative effects above opaque water clouds using a combination of A-Train satellite sensors, Atmos. Chem. Phys., 19, 4933-4962, https://doi.org/10.5194/acp-19-4933-2019, 2019. Kacenelenbogen, M., et al. "An evaluation of CALIOP/CALIPSO's aerosol‐above‐cloud detection and retrieval capability over North America." Journal of Geophysical Research: Atmospheres 119.1 (2014): 230-244. Liu, Z., Winker, D., Omar, A., Vaughan, M., Kar, J., Trepte, C., Hu, Y., and Schuster, G.: Evaluation of CALIOP 532 nm aerosol optical depth over opaque water clouds, Atmos. Chem. Phys., 15, 1265–1288, https://doi.org/10.5194/acp-15-1265-2015, 2015. Pistone, Kristina, et al. "Intercomparison of biomass burning aerosol optical properties from in-situ and remote-sensing instruments in ORACLES-2016."

---

## Referee Comment (RC3) · Anonymous Referee #3 · 6 May 2019

This paper provides a study of aerosols and clouds over the southeast Atlantic Ocean during the northern hemisphere summer season when smoke aerosols are transported over low-level stratocumulus clouds. The study is largely complementary to prior studies of the area. The unique contributions are the presentation of POLDER cloud properties in relation to the POLDER-retrieved aerosol optical thickness above the clouds and a radiative transfer model analysis constrained by observations and model reanalysis products to separate the contributions of aerosols and water vapor to changes in the radiative flux profiles during periods of high smoke concentration over the ocean.

The study largely confirms characteristics of the region previously described in the literature and, for the most part, supports prior hypotheses for how clouds respond to periods of high smoke transport in the layer above the clouds. The authors mount a hypothesis that variations in tropospheric humidity impact clouds through a weakening of cloud-top cooling. The paper may be suitable for publication in ACP, however I feel some the physical reasoning offered to support the authors' hypothesis for the impact on clouds of humidity variations requires a bit more rigor in its description, and I am concerned that it may rest on variations in a model-derived humidity profile that does not adequately resolve the vertical distribution of moisture to support the argument. Further comments on these and some other minor matters follows.

Major comments:

Page 13, lines 29-30 the authors claim that a difference in humidity at 925 hPa can explain the differences in LWP between the high and low AOT cases, but they do not explicitly describe the mechanism. Is the 925 hPa layer within the cloud layer or boundary layer where greater humidity is therefore able to condense, or is the 925 hPa level above the clouds and the authors are referring to a different mechanism? The physics behind this conclusion needs to be explained here.

Related to the previous comment, the physical reasoning described in the first two paragraphs of section 4.4 is difficult to follow. As mentioned above, the interpretation would seem to depend on whether the greater humidity at 925 hPa is considered in the cloud layer or not. Is it possible that the cloud-top pressure is sometimes below and sometimes above the 925 hPa level? If that is the case then there could be an artifact that appears as a difference between the high and low AOT cases, especially if the cloud-top height and cloud thickness is different between the two groups.

Also related to this is a concern about whether the ERA reanalysis is capturing the altitude and narrow thickness of the inversion layer at the top of the boundary layer. Is there some confidence that the inversion height is properly located in the vertical and

that the 925 hPa humidity in ERA corresponds well with observed humidity?

Minor comments:

Some of the imager-based cloud products from satellite sensors assume that clouds are plane-parallel and homogenous within the field of view of the instrument. Are the retrievals shown in figure 2 and discussed on page 9 lines 7-22 based on a similar assumption? Often the clouds over the southeast Atlantic Ocean are broken or otherwise horizontally heterogeneous at scales smaller than satellite footprints. If this is a source of uncertainty for the POLDER retrievals, it should be discussed here.

In sections 3.1 and 3.2 it talks about justification for the sampling area and time period, but arbitrarily sets the AOT thresholds that define "low" and "high". How are these values selected? And how many samples reside in the space in between where AOT is between 0.01 and 0.04?

Section 3.3 is titled "covariance between humidity and aerosol loading", but in the discussion the word "correlation" is used several times. On line 9 of page 11 it is even described as a "strong correlation". Nevertheless, there is no correlation analysis shown in this paper. Certainly, the word "strong" should not be used without actually evaluating a correlation coefficient and presenting it as such. I would recommend avoiding the word "correlation" here unless the correlation coefficient is evaluated and reported in the paper. A high/low grouping analysis can show statistically significant differences in a property even if the correlation coefficient between the grouping property (AOT in this case) and the other observed property (humidity) is low.

Page 12 lines 3-5 discusses changes in subsidence that are expected with smoke aerosol loading, but it is not explained why they are expected. What is the physical reasoning for a relationship between the smoke loading the environmental subsidence?

In section 4.2 there is discussion of the radiative fluxes and the it appears to me that the values are instantaneous values for the afternoon overpass time of the satellite. I think

it is important to clarify if the radiative fluxes correspond to mid-day values because in other papers values are reported as estimates of diurnal mean radiative fluxes.

Page 4, line 16: The Sakaeda et al. study used the global atmospheric model (CAM) coupled to a slab ocean. This is a coarse resolution model, not a large-eddy model, which usually refers to models that resolve some cloud-scale dynamics, which CAM does not.

---

## Author Comment (AC1) · 24 Jul 2019

**Author's response**

Journal: ACP
Title: Satellite inference of water vapor and aerosol-above-cloud combined effect on radiative budget and cloud top processes in the Southeast Atlantic Ocean
Author(s): Lucia T. Deaconu et al.
MS No.: acp-2019-189

Authors want to thank Referee #1, for his / her contribution and interactive comments. The answers to specific questions are addressed below, while the modifications made in the manuscript are in red.

**Anonymous Referee #1**

This paper presents an analysis of the variation and covariation of aerosol and cloud properties, as well as meteorological conditions, in the SE Atlantic region based on satellite observations. Compared to many previous studies on this topic, this paper sheds an important light on the co-variation of water vapor and above-cloud aerosols, and the implications for radiative effects on cloud. Overall, this paper is well written and well-organized. The topic is a good fit to ACP. On the other hand, it can be further improved if the following questions and issues can be addressed and clarified.

Comments/questions/suggestions:

1. A few important papers on the above-cloud aerosols in the SE Atlantic region are missing from the Introduction. They should be cited.
   - Page 2 Line 7~8: Swap et al. (1996) is perhaps the first one documenting the long-range transport of smoke aerosol from African continent to Atlantic.
   - Page 2, line 16: Recently, Zhang et al. (2016) provide a nice analysis of how DRE of above-cloud aerosol depends on AOT and COT (see their Figure 9).
   - Page 2, line 25: A few recent papers have studied the Twomey effect of above-cloud aerosols in the SE Atlantic region (Costantino and Bréon 2013; Lu et al. 2018). In particular, Lu et al. (2018), showed that the brightening effect due to entrained aerosols is actually stronger than the DRE and semi-direct effect.
   - Page 4, line 5~15: A recent study by Zhou et al. (2017) provides a more comprehensive picture of the radiative interactions between above-cloud aerosols and low clouds in the SE region.

Thank you for the suggestions.

2. Section 2.1: The Single-scattering albedo (SSA) is used later for deriving the absorption AOT. Is SSA a retrieved parameter or an a-priori assumption in the retrieval?    √

The SSA is parameter retrieved by POLDER, pixel by pixel (6x6 km$^2$) when the COT given by MODIS is larger than 3.0., therefore it varies both spatially and temporally. The details can be read in Peers et al., 2015.

3. POLDER retrievals are done at the 6x6 km. At this scale, a significant fraction of pixels would be partly cloudy. How are partly cloudy pixels treated in the analysis? If they are simply screened out, would that lead to sampling bias? MODIS Reff retrieval is not really biased when above-cloud aerosol is present. But MODIS COT retrieval can be substantially biased (underestimated). See Meyer et al. (2013). It needs to be clarified how the COT retrieval bias is accounted for the LWP estimation.  √

The POLDER retrievals at 6x6 km$^2$ use the 1x1 km$^2$ MODIS cloud fraction and Reff and COD as mask. All the MODIS pixels with cloud fraction lower than 1 and COD lower than 3 are not considered. We cite here Waquet et al., 2009: "We also used a cloud screening criterion to ensure that our analysis was only applied to cloudy pixels associated with an overcast cloud cover and a high cloud optical thickness (to ensure that the polarization cloud signal is saturated). The cloud screen took advantage of the high spatial resolution retrieval capabilities of MODIS (1x1km$^2$ at nadir) to estimate within each PARASOL pixel (6 x 6km$^2$) a mean value and a standard deviation for both the cloud optical thickness and the cloud droplet effective radius. In the analysis presented here, we only keep the PARASOL pixels associated with a mean cloud optical thickness larger than 5, a cloud optical thickness standard deviation smaller than 4, and a cloud droplet effective radius standard deviation smaller than 2."

In our study we compute the LWP using POLDER COT, which is corrected for the presence of absorbing aerosols above clouds. Indeed, the SSA and COT are retrieved simultaneously, and all details and uncertainties are described in Peers et al, 2015.

4. Page 7, Line 25: Indeed, CALIOP retrieval of above-cloud smoke AOT can be significantly biased, as pointed out by the authors and many other previous studies. This is because CALIOP cannot "see" the low portion of a thick smoke layer, i.e., it puts the bottom of aerosol layer too high. Therefore, simply using the extinction profile for HR computation seems to be a bad idea. It needs to be clarified here why this is justified.  √

We agree with the reviewer's comment that CALIOP underestimates the aerosol base altitude. However, this is the only available product to describe the vertical profile of aerosol above clouds from satellite. An alternative approach would consist in using the CALIOP data at 1064 nm, but this requires the development of a new CALIOP algorithm, which is out of the scope of this paper.

The aerosol layer base is generally high biased by approximately 500 m (see Rajapakshe et al., 2017). We recall that the CALIOP AOD at 532 nm is rescaled using POLDER AOD above clouds. Which means that the calculation of the heating rates is confined within a geometrical aerosol layer slightly thinner (by 500 m). The heating caused by the aerosols could occur further lower in the atmosphere (see Figure 11 – instead of a layer situated between 2.5 and 4.5 km we would go at 2 and 4.5 km), which doesn't change much the conclusions of our study. In the analysed area, aerosols are **mostly detached** from the cloud below, even considering the 500 m bias. Indeed, aerosol-cloud interaction is possible, but from our data, most of the retrievals

show detached situations. It is true that uncertainty in the aerosol layer height propagates to an uncertainty on the heating rate profile: the maximum of heating could be lower in the aerosol layer and the SW heating might be slightly closer to the cloud top (for example in Figure 11b, the mean HR_SW might not reach 10 K/day between the altitudes 3 and 4 km, and it might be higher around 2 km).

5. Page 8 Line 30: Again, is SSA a retrieved parameter or a pre-assumption. Is it a constant or it can vary spatially or temporally, and how?  √

Please see the answer to question 2.

6. Page 9 Line 10: It is interesting that the COT bias is pointed out here, but not in Section 2.1.  √

We have modified the text accordingly.

7. Page 13, The analysis in this section is somewhat superficial. Overall, it remains unclear after reading if those differences between low and high AOT loading cases in Figure 9 and 10 are significant and meaningful. Are they simply coincidence or there are some underlying physics? Significant revision is needed here. In particular, these differences should be put in the context of previous studies, e.g., Lu et al. 2018; Zhou et al. 2017  √

We thank the reviewer for pointing out the lack of clarity in section 4.1, as it is one piece of our demonstration of aerosol above cloud effects. In this section we show the difference in cloud properties between low and high aerosol-above-cloud loading, and it gives results consistent with Wilcox, 2010: there is a coincidence between the increase of aerosol loading and increase of liquid water content. The scope is to analyse and understand this coincidence. By choosing a time period and a small region, we have reduced the variability of meteorological parameters and thus its potential effects that could explain differences in cloud properties.

Firstly, we showed a significant difference in cloud water content: LWP increase systematically with 20 g/m$^2$ between the low and high polluted cases, from East to West over the zone. This increase is significant as it represents between 33% an 66%, from West to East, over the zone: +20g/m2 is +66% on the East and 33 % on the West.

What can be noticed is a difference in water vapor amounts over the zone (not very surprising after the results of Section 3.3) and in the covariance water vapor – LWP (Figure 10c). This difference in the covariance can be interpreted as a result of the reduction of negative feedbacks that are ordinarily at play for stratocumulus cloud field (Wood, 2002). One explanation could be due to difference in cloud top processes (forced by radiative effects) between low and high smoke loading. The rest of the paper presents the importance of smoke-over-cloud radiative effect and the estimate of water vapor and aerosol distinct contributions. Another explanation for Figure 10c is the presence of precipitable clouds in the low case scenario, which is supported by the existence of larger cloud droplet effective radii (> 11 μm, Figure 9a) associated to large specific humidity between 8° and 6° E (Figure 9d). These could lead to the shift in LWP observed in Figure 10c, where LWP decreases abruptly for $q_v$ larger than 9g/kg. Therefore, the difference observed between LWP-$q_v$ for the high and

low situations might be explained by two hypotheses: the radiative effect of smoke layer on the underlaying cloud or the loss of precipitable water. We modified the text.

8. Page 14 Line 4~7: It remains puzzling to me why Figure 10c "seems well illustrating the cloud-radiation-entrainment feedback". It needs to be explained better and, in more detail, here. What is the underlying physics of this feedback? Why does the decrease of LWP with qv "illustrate" this feedback?  √

Thank you for your comment. Please refer to the previous comment and answer.
In Discussion section 4.4 we interpret this figure, making the link with the study of the smoke radiative forcing and the potential forcing of cloud top processes and feedbacks, while discussing the physical link of LWP with qv at 925 hPa, depending on the location of this pressure level compared with the cloud top altitude. We have also considered a second hypothesis that could explain this behaviour.

9. Page 15 Line 5: The total HR at cloud top for the unpolluted case is -13.5 K/day and the corresponding value for polluted cases is -12.9 K/day. Considering that the LW cooling rate for both groups is the same (i.e., -17.79 K/day). This difference indicates that the SW heating at the cloud top for the polluted case is actually stronger (i.e., more positive). Isn't this counterinitiative? Shouldn't the extinction of above-cloud aerosol layer reduce the SW heating rate at cloud top in the polluted cases? This SW heating difference should be pointed out here and explained in detail.  √

Thank you for pointing this out. The SW heating at the top of the cloud in figure 11 was a labelling mistake. The correct values show a cooling of 0.56 K/day at the cloud top due to the presence of aerosols above. We corrected the figures and the text.

10. The overall values between the polluted and unpolluted cases in Figure 11 seem pretty close. Some statistical test (e.g., T-test) need to be performed to tell if the differences are statistically significant.  √

In figure 11 we were interested in the broad difference between an unpolluted atmospheric profile (that contains only water vapor) and one that also includes an aerosol layer above the cloud. For all the CALIOP extinction profiles with an aerosol layer above clouds we selected the associated water vapor and temperature profiles and computed the heating rates by turning on and off the aerosols above the clouds in the model. The 6 K/day warming between 2.5 and 4.5 km due to the aerosols is clearly showing the effect of aerosols. Indeed, at the cloud top the effect of both aerosol and water vapor is less obvious, due to the different cloud altitudes retrieved from CALIOP that position the strong cloud-top cooling associated to LW cooling at different altitudes. This is obvious from the large values of the standard deviation resulted after averaging these profiles (horizontal bars in Figure 11). This is another reason why for the next part of the study we have analysed a single profile. A t-test would show significant differences at the aerosol layer, but almost no statistical difference at the cloud top, which is what we expect from this figure.

- Costantino, L., and F. M. Bréon (2013), Aerosol indirect effect on warm clouds over South-East Atlantic, from co-located MODIS and CALIPSO observations, Atmospheric Chemistry and Physics, 13(1), 69–88, doi:10.5194/acp-1369-2013.
- Lu, Z., X. Liu, Z. Zhang, C. Zhao, K. Meyer, C. Rajapakshe, C. Wu, Z. Yang, and J. E. Penner (2018), Biomass smoke from southern Africa can significantly enhance the brightness of stratocumulus over the southeastern Atlantic Ocean, PNAS, 115(12), 201713703–2929, doi:10.1073/pnas.1713703115.
- Meyer, K., S. Platnick, L. Oreopoulos, and D. Lee (2013), Estimating the direct radiative effect of absorbing aerosols overlying marine boundary layer clouds in the southeast Atlantic using MODIS and CALIOP, Journal of Geophysical Research-Atmospheres, 118(10), 4801–4815, doi:10.1002/jgrd.50449.
- Swap, R., M. Garstang, S. A. Macko, P. D. Tyson, W. Maenhaut, P. Artaxo, P. Kållberg, and R. Talbot (1996), The long‰range transport of southern African aerosols to the tropical South Atlantic, J. Geophys. Res., 101(D19), 23777–23791, doi:10.1029/95JD01049.
- Zhang, Z., K. Meyer, H. Yu, S. Platnick, P. Colarco, Z. Liu, and L. Oreopoulos (2016), Shortwave direct radiative effects of above-cloud aerosols over global oceans derived from 8 years of CALIOP and MODIS observations, Atmospheric Chemistry and Physics, 16(5), 2877–2900, doi:10.5194/acp-16-2877-2016.
- Zhou, X., A. S. Ackerman, A. M. Fridlind, R. Wood, and P. Kollias (2017), Impacts of solar-absorbing aerosol layers on the transition of stratocumulus to trade cumulus clouds, Atmospheric Chemistry and Physics, 17(20), 12725–12742, doi:10.5194/acp17-12725-2017.

Authors want to thank Referee #2, Meloe Kacenelenbogen for her contribution and interactive comments. The answers to specific questions are addressed below, while the modifications made in the manuscript are in red.

**Meloe Kacenelenbogen (Referee)**
meloe.s.kacenelenbogen@nasa.gov

The authors use a combination of POLDER, MODIS, CALIOP and modelled meteorological profiles to evaluate the changes in met. parameters (e.g., Temperature, RH, Specific Humidity, Winds), cloud properties (droplet effective radius, top height, liquid water path), and heating rates as a function of more or less overlying AAOD. This paper is of good quality, well written and structured. It will be worthy of publication, once the issues below are addressed.

Overall comments:
1. Section 2.1. could benefit from a Table listing all the products and corresponding satellites/ models used in their method.    √

Thank you. We have added a table summarizing all the satellite and models, with the parameters used in this study.

2. The authors base their study near the coast because this is where "aerosols are mainly detached from clouds" using CALIOP (by the way, CALIOP will likely miss the base of the aerosols). But then, further in their study, they analyze potential aerosol-cloud interactions. It would be worth adding some information on aerosol-cloud contact frequency over the region

In this study we are not analysing the aerosol/cloud interaction due to physical contact (first indirect effect). Instead, we focus on the "semi-direct effect" (more specifically: we study the changes in the radiative budget above cloud and at cloud top and their implications for the cloud dynamical development).
Regarding the aerosol-cloud contact frequency in this region, please refer to Figure 13 of Constantino and Breon, 2013 (ACP), where they show the number concentration of coincident MODIS-CALIOP retrievals for mixed aerosol-cloud layer and aerosols above clouds. Based on their results, we notice that our small region of interest is associated to few contact (or mixed) situations. Indeed, aerosol-cloud interaction is possible, but from our data, most of the retrievals show detached situations

[Figure]

Number of retrievals within a 2x2 degree box

Mixed cloud and aerosol layers

Aerosol above cloud layer

**Fig. 13.** Number concentration of coincident MODIS-CALIPSO retrievals for all aerosol regimes, in the region within [2S–15S; 14W–18E]. Colour scale represents number of measurements within a 2 × 2 degree box, for cases of mixed (left image) and well separated (right image) cloud-aerosol layers.

3. The reader could benefit from an explanation of their AAOD thresholds (i.e., >0.01 and <0.04); If AAOD is the threshold, and it says "high" or "low" aerosol loading, does this mean that the authors assume a constant SSA value? If not, shouldn't they say "higher loading and higher absorption" instead?    √

These values are empirically chosen. The strategy behind the choice is: with AAOD > 0.04, we select the cases where the radiative impact of the smoke layer is expected to be maximized (and conversely minimized for < 0.01) to better highlight the potential semi-effect of smoke layer on the below cloud. The SSA is not constant, it is retrieved by POLDER at 6x6 km² spatial resolution (see Peers et al., 2015 ACP).

If not, shouldn't they say "higher loading and higher absorption" instead?
Thank you, we modified the text.

Detailed comments: .

4. I had to read the title multiple times to understand it. "Combined effects of water vapor and aerosols on underlying cloud top processes and radiative budget from satellites over the South East Atlantic Ocean" or something along those lines would make it clearer.    √

Thank you, but we prefer our current title.

5. P1, line 14: "it is a prerequisite" .    √
6. P1, line 21: "sensing techniques" .    √
7. P2, Line 10: "negligible wet scavenging" needs more references.

We are referring to wet scavenging of aerosols above stratocumulus clouds, without overlaid cirrus clouds, which by definition is negligible.

8. P3, line 11, line 30 (and other places): should read "cloud properties", "particle size", "droplet effective radius" etc.. √
9. P3, line 20: I suggest briefly describing the "assumptions" (i.e., mostly the CALIOP lidar ratio)  √

10. P3, line 21: when introducing the depolarization method, I suggest saying "first introduced by Hu et al., 2007 and further implemented by e.g., Liu et al., 2015, Kacenelenbogen et al., 2019 Deaconu et al., 2017" . √
11. P3, L22: "AAC properties" . √
12. P3, L31: I suggest mentioning SSA from Peers et al.(2015) is retrieved above clouds. √
13. P4, L3: please consider referring to Table 1 or 2 of Kacenelenbogen et al. [2019] . √
14. P4, L15: "high loadings of smoke" . √
15. P5, L14: I suggest describing the "semi-direct effect" . √ We modified the text
16. P5, L30: available at 490 and 865 nm . √
17. P5, L31: I suggest briefly describing the Angstrom exponent . √
18. P5, L32: "the aerosol model prescribed in the POLDER (?) satellite algorithm" . √
19. P6, L6: Are MODIS cloud properties corrected for AAC? . √

The MODIS cloud effective radius is not corrected for absorption, but the bias due to absorbing aerosol above clouds on the MODIS Reff retrievals are of about 2 %. We do not use MODIS COD in our study. The POLDER COT is indeed corrected for absorption.

20. P6, L7: "cloud altitude derived from POLDER (ZO2)" . √
21. P6, L7: "ZO2 is calculated using. . ." . √
22. P6, Section 2.1: As said in the overall comments, this section could benefit from a Table listing all the products and corresponding satellites/ models used in the method. . √ We added a table in the study
23. P7, L3: "The GAME model" . √
24. P7, L4: Instead of "for this", I suggest "inputs to GAME are. . ." . √
25. P7, L10: Instead of "we", I suggest "GAME uses" . √
26. P7, L12: Instead of "the CALIOP method", I suggest "the standard CALIOP product can underestimate. . ." And this is happening also when aerosols are below a certain detection threshold. You could reference Kacenelenbogen et al. [2014]. . √
27. P7, L24: "using the POLDER method" . √
28. P7, L25: I suggest mentioning that, although this might not affect your study, the aerosol base height might still very likely be biased high after scaling the extinction profile as seen on Fig. 1 . √ We modified the text
29. P8, L7: Consider replacing easterlies by easterly winds . √
30. P8, L27: I suggest "decrease" instead of "go down" . √
31. P8, L28: I suggest "value prescribed for the dust model in the POLDER algorithm" if I understand this correctly. √.
32. P9, L1: For SSA values during ORACLES-2016, I suggest referencing Pistone et al., [2019] . √
33. P9, L13: I suggest describing Fig. 2g before Fig. 3 . √
34. P9, L17: "lower than" . √
35. P9, L19: "Nevertheless, the stratocumuli are low-level clouds, so, an underestimation of around 300 m by the POLDER product is more likely": this is not clear to me. . √

Oxygen pressure technique developed for POLDER tends to slightly underestimate the cloud top altitude in cases of low-level clouds underlying by BBA (first observed in Waquet et al, 2009 and now quantified in Figure 3 with more data)

36. P9, L21: "stratocumulus become more fractioned" would it be worth showing the cloud fraction as well? . √

We meant that stratocumulus change into cumulus. We modified the text

37. P10, L9: I suggest explaining the thresholds of 0.01 and 0.04 for AAOD. Could it be an AOD of 0.2 at 865nm with an SSA of 0.8? . √

Please refer to the answer given at question 3

38. P10, L17: "aerosols are mainly detached from low level clouds": I suggest to be more quantitative. Again, CALIOP aerosol base height is biased high. There is likely more contact than what is seen from CALIOP. Maybe use results from Rajapakshe et al. [2017]? . √

We agree with the reviewer's comment that CALIOP underestimates the aerosol base altitude. The aerosol layer base is generally high biased by approximately 500 m (Rajapakshe et al., 2017). The CALIOP AOD at 532 nm is rescaled using POLDER AOD above clouds. Which means that the calculation of the heating rates is confined within a geometrical aerosol layer slightly thinner (by 500 m). The heating caused by the aerosols could occur further lower in the atmosphere (see Figure 11 – instead of a layer situated between 2.5 and 4.5 km we would go at 2 and 4.5 km), which doesn't change much the conclusions of our study. In the analysed area, aerosols are mostly detached from the cloud below, even considering the 500 m bias.

39. P10, L31: "June to August (JJA) 2008" . √
40. P11, L7: "from 7.5 to 10 g.kg-1" . √
41. P11, L8: "smoke plume level (i.e., between 850 and 700 hPa)" . √
42. P11, L18: "plumes resulting from" . √
43. P11, L30: "Figure 7b and 7c" . √
44. P12, L1: I suggest "smoke" instead of "polluted" (urban pollution and smoke being two different aerosol types when using satellite remote sensing) . √
45. P12, L25: the authors choose the sampling area so that "aerosols are mainly detached" from clouds. Are you implying more aerosol-cloud contact here? . √

We believe the reviewer misunderstood this sentence. "Mainly detached" stands for less aerosol-cloud contact situations. Figure 4 shows PDFs of aerosol base and top altitudes and cloud top altitudes retrieved from CALIOP along bins of longitude, over the entire SOA region. Between May and July, we can notice less contact between 13° and 9° E - average aerosol base altitude is 2.21±0.84 km, while the average cloud top altitude is 0.84±0.3 km - and 9° and 5° E - average aerosol base altitude is 1.85±0.75 km, while the average cloud top altitude is 1.14±0.32 km. In August – October period the aerosol layers even more elevated from the clouds. As discussed previously, these values may be biased by 500 m, but we can expect more detached situations closer to the coast. Here is a figure presenting the mean per bin of longitude of the cloud top altitude, as well as aerosol base and top altitudes retrieved from CALIOP, for JJA over the region of interest. Please also notice the standard deviation vertical bars for cloud top altitude and aerosol base altitude. Indeed, aerosol-cloud interaction is possible, but from our data, most of the retrievals show detached situations. We added this figure in the paper.

[Figure]

*Figure X: Mean cloud top altitude, aerosol base altitude and aerosol top altitude as a function of longitude, over the sample region for June-August 2008. Vertical bars represent the standard deviation*

46. P 13, L1: This is not clear. I would rephrase. . √ We modified the text
47. P13, L14: I would quantify how low the difference is. .         √
48. P13, L20: "wind speed (see figure 10)" .        √
49. P17, L12: "South East Atlantic Ocean" . √
50. P17, L13: "increase in size, decrease in absorption" . √
51. P17, L20: why "advanced"?; replace "forcing" by "effects" . √
52. P17, L21: I would quantify "lower" . √
53. Fig. 4: I suggest adding that these results use CALIOP data in the legend; First row could say MJJ and second row could say ASO .         √
54. Fig. 5, 6, 7: An illustration of the mean aerosol and cloud layer heights for the sampling domain and period would help the reader . √ We added the figure as a supplement
55. Fig. 7d: Legend is confusing: is it now blue for august and red for June-July? .         √
56. Fig. 9: It would not hurt to remind the reader that this is about clouds only and add "at 925hPa" to the y-axis of Fig. 9d . √
57. Fig. 7, 9, 10: I find it non-intuitive to color the "low" aerosol loading conditions in red and the "high" aerosol loading conditions in blue. I would have done the opposite. √

Thank you for the good suggestion.

- Deaconu, L. T., Waquet, F., Josset, D., Ferlay, N., Peers, F., Thieuleux, F., Ducos, F., Pascal, N., Tanré, D., Pelon, J., and Goloub, P.: Consistency of aerosols above clouds characterization from A-Train active and passive measurements, Atmos. Meas. Tech., 10, 3499–3523, https://doi.org/10.5194/amt-10-3499-2017, 2017.
- Kacenelenbogen, M. S., Vaughan, M. A., Redemann, J., Young, S. A., Liu, Z., Hu, Y., Omar, A. H., LeBlanc, S., Shinozuka, Y., Livingston, J., Zhang, Q., and Powell, K. A.: Estimations of global shortwave direct aerosol radiative effects above opaque water clouds using a combination of A-Train satellite sensors, Atmos. Chem. Phys., 19, 4933-4962, https://doi.org/10.5194/acp-19-4933-2019, 2019.

- Kacenelenbogen, M., et al. "An evaluation of CALIOP/CALIPSO's aerosol above cloud detection and retrieval capability over North America." Journal of Geophysical Research: Atmospheres 119.1 (2014): 230-244.
- Liu, Z., Winker, D., Omar, A., Vaughan, M., Kar, J., Trepte, C., Hu, Y., and Schuster, G.: Evaluation of CALIOP 532 nm aerosol optical depth over opaque water clouds, Atmos. Chem. Phys., 15, 1265–1288, https://doi.org/10.5194/acp-151265-2015, 2015.
- Pistone, Kristina, et al. "Intercomparison of biomass burning aerosol optical properties from in-situ and remote-sensing instruments in ORACLES-2016." Interactive comment on Atmos. Chem. Phys. Discuss., https://doi.org/10.5194/acp-2019-189,2019.

Authors want to thank Referee #2 for his / her contribution and interactive comments. The answers to specific questions are addressed below, while the modifications made in the manuscript are in red.

**Anonymous Referee #3**

This paper provides a study of aerosols and clouds over the southeast Atlantic Ocean during the northern hemisphere summer season when smoke aerosols are transported over low-level stratocumulus clouds. The study is largely complementary to prior studies of the area. The unique contributions are the presentation of POLDER cloud properties in relation to the POLDER-retrieved aerosol optical thickness above the clouds and a radiative transfer model analysis constrained by observations and model reanalysis products to separate the contributions of aerosols and water vapor to changes in the radiative flux profiles during periods of high smoke concentration over the ocean.

The study largely confirms characteristics of the region previously described in the literature and, for the most part, supports prior hypotheses for how clouds respond to periods of high smoke transport in the layer above the clouds. The authors mount a hypothesis that variations in tropospheric humidity impact clouds through a weakening of cloud-top cooling. The paper may be suitable for publication in ACP, however I feel some the physical reasoning offered to support the authors' hypothesis for the impact on clouds of humidity variations requires a bit more rigor in its description, and I am concerned that it may rest on variations in a model-derived humidity profile that does not adequately resolve the vertical distribution of moisture to support the argument. Further comments on these and some other minor matters follows.

Major comments:
1. Page 13, lines 29-30 the authors claim that a difference in humidity at 925 hPa can explain the differences in LWP between the high and low AOT cases, but they do not explicitly describe the mechanism. Is the 925 hPa layer within the cloud layer or boundary layer where greater humidity is therefore able to condense, or is the 925 hPa level above the clouds and the authors are referring to a different mechanism? The physics behind this conclusion needs to be explained here.  √

The authors agree with the reviewer's comment. From the ECMWF reanalysis data, the model-derived humidity profile has an uncertainty. Also, from this estimated profile, it is not really obvious that the 925 hPa level is every time within the stratocumulus layer or just above it. From the analysis of model-derived humidity profile over the zone, values of specific humidity between 8 to 12 g/kg would correspond to humidity within the cloud layers. Values for layers just above the cloud top would be mostly around 6g/kg, at the 900 hPa level. It is true that for the two cases (humidity above the cloud layers or within them), the covariance between humidity at 925 hPa and LWP does not have the same implication. From the analysis of model-derived humidity profile over the area, values of specific humidity between 8 to 12 g/kg would correspond to increased humidity within the cloud, hence more condensable water. Alternatively, if this humidity is present above the cloud, the entrainment process

would also lead to more condensed water in the cloud. Either way the increased humidity can lead to increased condensable water within the cloud layer. We modified the text in order to describe in a more careful way the figure that shows the covariance between humidity at 925 hPa and LWP (Figure 10c). We keep for later in the text (Discussion section 4.4) an interpretation of this figure, discussing the physical link of LWP with qv at 925 hPa, depending on the location of this pressure level compared with the cloud top altitude. We make the link with the study of the smoke radiative forcing and the potential forcing of cloud top processes and feedbacks. We have also considered a second hypothesis related to the loss of precipitable water that could explain this behaviour.

2. Related to the previous comment, the physical reasoning described in the first two paragraphs of section 4.4 is difficult to follow. As mentioned above, the interpretation would seem to depend on whether the greater humidity at 925 hPa is considered in the cloud layer or not. Is it possible that the cloud-top pressure is sometimes below and sometimes above the 925 hPa level? If that is the case then there could be an artefact that appears as a difference between the high and low AOT cases, especially if the cloud-top height and cloud thickness is different between the two groups.

Also related to this is a concern about whether the ERA reanalysis is capturing the altitude and narrow thickness of the inversion layer at the top of the boundary layer. Is there some confidence that the inversion height is properly located in the vertical and that the 925 hPa humidity in ERA corresponds well with observed humidity? √

Please refer to Adebiyi et al., 2015, Figures 13 and 14, in which they show temperature, relative humidity and wind profile at St. Helena island, in the South East Atlantic Ocean, from radiosonde measurements and reanalysis data. We cite here: "Averages over the terciles show that **the reanalyses largely capture the mean thermodynamical structure of the soundings** (Fig. 13), with the largest discrepancies between 700 and 1000 hPa, where the reanalyses with higher vertical resolutions (MERRA and ERA-Interim) perform better than NCEP, although all have difficulty representing the boundary layer-top inversions." From the below figure, we notice that ERA-Interim captures best the potential temperature profile compared to the radiosondes measurements, and shows good results for relative humidity and specific humidity. The wind components are less well captured by all the reanalyses data.

[Figure]

Minor comments:

3. Some of the imager-based cloud products from satellite sensors assume that clouds are plane-parallel and homogenous within the field of view of the instrument. Are the retrievals shown in figure 2 and discussed on page 9 lines 7-22 based on a similar assumption?   √

Yes, the clouds are considered plane-parallel and homogeneous within the field of view.

4. Often the clouds over the southeast Atlantic Ocean are broken or otherwise horizontally heterogeneous at scales smaller than satellite footprints. If this is a source of uncertainty for the POLDER retrievals, it should be discussed here. √

POLDER retrievals are only performed for homogeneous and optically thick cloud cover. The variability of the cloud properties is quantified using MODIS product retrieved at a finer resolution and the heterogeneous and broken-field clouds are removed from the POLDER retrievals. Various filters and criteria are used the improve POLDER above cloud products quality, we refer to Waquet et al., 2015 for details. Moreover, the 3D radiative impacts in the POLDER above cloud retrievals were studied in Waquet et al., 2013 (AMT) for the polarization data and in Peers et al., 2015 (ACP) for the total intensity measurements.

5. In sections 3.1 and 3.2 it talks about justification for the sampling area and time period, but arbitrarily sets the AOT thresholds that define "low" and "high". How are these values selected?

These values are empirically chosen. The strategy behind the choice is: with AAOD > 0.04, we select the cases where the radiative impact of the smoke layer is expected to be maximized (and conversely minimized for < 0.01) to better highlight the potential semi-effect of smoke layer on the below cloud.

6. Section 3.3 is titled "covariance between humidity and aerosol loading", but in the discussion the word "correlation" is used several times. On line 9 of page 11 it is even described as a "strong correlation". Nevertheless, there is no correlation analysis

shown in this paper. Certainly, the word "strong" should not be used without actually evaluating a correlation coefficient and presenting it as such. I would recommend avoiding the word "correlation" here unless the correlation coefficient is evaluated and reported in the paper. A high/low grouping analysis can show statistically significant differences in a property even if the correlation coefficient between the grouping property (AOT in this case) and the other observed property (humidity) is low.  √

We followed the reviewer's comment: we modified the text and replaced "correlation" with "covariance" on line 9 of Section 3.3, and everywhere in the manuscript.

7. Page 12 lines 3-5 discusses changes in subsidence that are expected with smoke aerosol loading, but it is not explained why they are expected. What is the physical reasoning for a relationship between the smoke loading the environmental subsidence?  √

The coincidence between smoke loading and change in subsidence is the result of large and regional scales circulation, as described by Adebiyi et al (2015). They described from ERA-Interim reanalysis a reduction of free-tropospheric subsidence when more absorbing aerosols are present, also seen in the model results of Sakaeda et al. (2011). We modified the text with: "... aerosols are present above the clouds (mainly during June-July). It confirms the description of Adebiyi et al. (2015), see their Figure 15, that the large-scale subsidence decreases when aerosol loadings are higher, which would tend to push the cloud to rise in altitude."

8. In section 4.2 there is discussion of the radiative fluxes and the it appears to me that the values are instantaneous values for the afternoon overpass time of the satellite. I think it is important to clarify if the radiative fluxes correspond to mid-day values because in other papers values are reported as estimates of diurnal mean radiative fluxes.  √

Indeed, the reported values are instantaneous, at the POLDER overpass hour (around 13:30 local time). We have also calculated the diurnal variation of the heating rate profile (K/day) in the shortwave and longwave spectrum, as well as their total (see figure below), using the mean values of the aerosol, cloud and meteorological parameters, but we haven't reported it in this paper. We noticed that in the shortwave spectrum the warming between 3 and 4.5 km increases from 2 K/day in the morning (at 6 h) to approximately 8 K/day around noon, and after 14 h the heating rate starts to decrease. Similar variation in shortwave spectrum is observed at the cloud top. In longwave we do not observe a dependence of the heating rate with the hour, because the water vapor longwave absorption/emission is not affected by the solar zenith angle. We notice more cooling (around -4 K/day) between 3.5 and 5.5 km due to the water vapor content in the atmospheric column, and a strong cooling at the cloud top. The scale is fixed between -8.0 and 0.0 K/day, but the true value goes down to -70 K/day. The total heating rate also varies during the day; the longwave cooling only partially compensates the shortwave heating between 8 h and 16 h. We observe a maximum heating of 5 K/day at noon at 4 km; early in the morning and late in the afternoon we notice, however, a total cooling of -1.7 K/day. The cloud top remains under strong cooling.

[Figure]

*Figure 1 Average diurnal variation of heating rate profiles calculated in shortwave spectrum (VIS), in longwave spectrum (IRT) and total heating rate, within the selected area.*

9. Page 4, line 16: The Sakaeda et al. study used the global atmospheric model (CAM) coupled to a slab ocean. This is a coarse resolution model, not a large-eddy model, which usually refers to models that resolve some cloud-scale dynamics, which CAM does not.   √

We modified the text and deleted "large-eddy".

---

## Author Comment (AC3) · 24 Jul 2019

Dear Sir/Mam,

Please find attached in the supplement a point-by-point answer from the authors. We have done our best to respond to your questions and to improve the paper. All co-authors have contributed to these answers and concur with the submission.

Yours sincerely,

[Figure]

Lucia Deaconu

Please also note the supplement to this comment:
https://www.atmos-chem-phys-discuss.net/acp-2019-189/acp-2019-189-AC3-supplement.pdf

---

## Author Comment (AC4) · 24 Jul 2019

[revised manuscript text omitted]

SSA 865 nm (and absorption AOT)
COT, corrected for absorption
Cloud altitude $ZO_2$ (km) | Waquet et al., 2009
Waquet et al., 2013
Peers et al., 2015
Vanbauce et al., 2003 |
| MODIS | $r_{eff}$ ($\mu$m) | Meyer et al., 2015 |
| CALIOP | $\sigma_{aer}$ at 532 nm (and AOT)
Aerosol base and top altitudes (km)
Cloud top altitudes (km) | Vaughan et al., 2009;
Young and Vaughan, 2009 |
| ECMWF | Temperature (K)
Relative humidity (%)
Specific humidity (g.kg$^{-1}$)
Wind direction (°)
Wind amplitude (m.s$^{-1}$) | Berrisford et al., 2011
Dee et al., 2011 |
| GAME radiative transfer model | Heating rates in SW and LW (Kday$^{-1}$)
DRE at TOA (Wm$^{-2}$) | Dubuisson et al., 2006 |

[revised manuscript text omitted]

Font color: Auto

| Page 13: [2] Formatted | Lucia Deaconu | 7/21/19 9:14:00 PM |
|---|---|---|

Font: Italic

| Page 13: [3] Formatted | Lucia Deaconu | 7/21/19 9:14:00 PM |
|---|---|---|

Font: Italic

| Page 13: [4] Formatted | Lucia Deaconu | 7/21/19 9:11:00 PM |
|---|---|---|

Font color: Auto

| Page 13: [5] Formatted | Lucia Deaconu | 7/21/19 9:11:00 PM |
|---|---|---|

Font: (Default) +Body (Times New Roman)

| Page 13: [6] Deleted | Lucia Deaconu | 7/22/19 11:30:00 AM |
|---|---|---|

| Page 13: [7] Deleted | Lucia Deaconu | 7/19/19 9:56:00 PM |
|---|---|---|

| Page 13: [8] Formatted | Lucia Deaconu | 7/22/19 2:30:00 PM |
|---|---|---|

Font: Italic

| Page 13: [9] Formatted | Lucia Deaconu | 7/22/19 2:30:00 PM |
|---|---|---|

Font: Italic, Subscript

| Page 13: [10] Formatted | Lucia Deaconu | 7/22/19 12:18:00 PM |
|---|---|---|

Font: 10 pt

| Page 13: [11] Formatted | Lucia Deaconu | 7/22/19 12:18:00 PM |
|---|---|---|

Font: 10 pt

| Page 13: [12] Formatted | Lucia Deaconu | 7/22/19 12:18:00 PM |
|---|---|---|

Font: 10 pt

| Page 13: [13] Formatted | Lucia Deaconu | 7/22/19 12:18:00 PM |
|---|---|---|

Font: 10 pt

| Page 13: [14] Formatted | Lucia Deaconu | 7/22/19 12:18:00 PM |
|---|---|---|

Font: 10 pt

| Page 13: [15] Formatted | Lucia Deaconu | 7/22/19 12:18:00 PM |
|---|---|---|

Font: 10 pt

| Page 13: [16] Formatted | Lucia Deaconu | 7/22/19 2:32:00 PM |
|---|---|---|

| Page 13: [17] Formatted | Lucia Deaconu | 7/22/19 2:40:00 PM |
|---|---|---|

Font: Italic

| Page 13: [18] Formatted | Lucia Deaconu | 7/22/19 2:40:00 PM |
|---|---|---|

Font: Italic

| Page 15: [19] Deleted | Lucia Deaconu | 7/23/19 2:56:00 PM |
|---|---|---|

| Page 15: [20] Formatted | Lucia Deaconu | 7/23/19 3:04:00 PM |
|---|---|---|

Font color: Auto

| Page 15: [20] Formatted | Lucia Deaconu | 7/23/19 3:04:00 PM |
|---|---|---|

Font color: Auto

| Page 15: [20] Formatted | Lucia Deaconu | 7/23/19 3:04:00 PM |
|---|---|---|

Font color: Auto

| Page 15: [20] Formatted | Lucia Deaconu | 7/23/19 3:04:00 PM |
|---|---|---|

Font color: Auto

| Page 15: [20] Formatted | Lucia Deaconu | 7/23/19 3:04:00 PM |
|---|---|---|

Font color: Auto

| Page 15: [20] Formatted | Lucia Deaconu | 7/23/19 3:04:00 PM |
|---|---|---|

Font color: Auto

| Page 15: [20] Formatted | Lucia Deaconu | 7/23/19 3:04:00 PM |
|---|---|---|

Font color: Auto

| Page 15: [20] Formatted | Lucia Deaconu | 7/23/19 3:04:00 PM |
|---|---|---|

Font color: Auto

| Page 15: [20] Formatted | Lucia Deaconu | 7/23/19 3:04:00 PM |
|---|---|---|

Font color: Auto

| Page 15: [20] Formatted | Lucia Deaconu | 7/23/19 3:04:00 PM |
|---|---|---|

Font color: Auto

| Page 15: [20] Formatted | Lucia Deaconu | 7/23/19 3:04:00 PM |
|---|---|---|

Font color: Auto

| Page 15: [20] Formatted | Lucia Deaconu | 7/23/19 3:04:00 PM |
|---|---|---|

Font color: Auto

**Page 15: [20] Formatted** | **Lucia Deaconu** | **7/23/19 3:04:00 PM**

Font color: Auto

**Page 15: [20] Formatted** | **Lucia Deaconu** | **7/23/19 3:04:00 PM**

Font color: Auto

**Page 15: [20] Formatted** | **Lucia Deaconu** | **7/23/19 3:04:00 PM**

Font color: Auto

**Page 15: [20] Formatted** | **Lucia Deaconu** | **7/23/19 3:04:00 PM**

Font color: Auto

**Page 15: [20] Formatted** | **Lucia Deaconu** | **7/23/19 3:04:00 PM**

Font color: Auto

**Page 15: [20] Formatted** | **Lucia Deaconu** | **7/23/19 3:04:00 PM**

Font color: Auto

**Page 15: [20] Formatted** | **Lucia Deaconu** | **7/23/19 3:04:00 PM**

Font color: Auto

**Page 15: [20] Formatted** | **Lucia Deaconu** | **7/23/19 3:04:00 PM**

Font color: Auto

**Page 15: [20] Formatted** | **Lucia Deaconu** | **7/23/19 3:04:00 PM**

Font color: Auto

**Page 15: [20] Formatted** | **Lucia Deaconu** | **7/23/19 3:04:00 PM**

Font color: Auto

**Page 15: [20] Formatted** | **Lucia Deaconu** | **7/23/19 3:04:00 PM**

Font color: Auto

**Page 15: [20] Formatted** | **Lucia Deaconu** | **7/23/19 3:04:00 PM**

Font color: Auto

**Page 15: [20] Formatted** | **Lucia Deaconu** | **7/23/19 3:04:00 PM**

Font color: Auto

**Page 15: [20] Formatted** | **Lucia Deaconu** | **7/23/19 3:04:00 PM**

Font color: Auto

**Page 15: [20] Formatted** | **Lucia Deaconu** | **7/23/19 3:04:00 PM**

| Page 15: [20] Formatted | Lucia Deaconu | 7/23/19 3:04:00 PM |
|---|---|---|

Font color: Auto

| Page 15: [20] Formatted | Lucia Deaconu | 7/23/19 3:04:00 PM |
|---|---|---|

Font color: Auto

| Page 15: [20] Formatted | Lucia Deaconu | 7/23/19 3:04:00 PM |
|---|---|---|

Font color: Auto

| Page 15: [20] Formatted | Lucia Deaconu | 7/23/19 3:04:00 PM |
|---|---|---|

Font color: Auto

| Page 15: [20] Formatted | Lucia Deaconu | 7/23/19 3:04:00 PM |
|---|---|---|

Font color: Auto

| Page 15: [20] Formatted | Lucia Deaconu | 7/23/19 3:04:00 PM |
|---|---|---|

Font color: Auto

| Page 15: [20] Formatted | Lucia Deaconu | 7/23/19 3:04:00 PM |
|---|---|---|

Font color: Auto

| Page 15: [20] Formatted | Lucia Deaconu | 7/23/19 3:04:00 PM |
|---|---|---|

Font color: Auto

| Page 15: [20] Formatted | Lucia Deaconu | 7/23/19 3:04:00 PM |
|---|---|---|

Font color: Auto

| Page 15: [20] Formatted | Lucia Deaconu | 7/23/19 3:04:00 PM |
|---|---|---|

Font color: Auto

| Page 15: [20] Formatted | Lucia Deaconu | 7/23/19 3:04:00 PM |
|---|---|---|

Font color: Auto

| Page 15: [21] Formatted | Lucia Deaconu | 7/23/19 3:04:00 PM |
|---|---|---|

Font color: Auto, Not Superscript/ Subscript

| Page 15: [21] Formatted | Lucia Deaconu | 7/23/19 3:04:00 PM |
|---|---|---|

Font color: Auto, Not Superscript/ Subscript

| Page 15: [21] Formatted | Lucia Deaconu | 7/23/19 3:04:00 PM |
|---|---|---|

Font color: Auto, Not Superscript/ Subscript

| Page 15: [21] Formatted | Lucia Deaconu | 7/23/19 3:04:00 PM |
|---|---|---|

Font color: Auto, Not Superscript/ Subscript

| Page 15: [21] Formatted | Lucia Deaconu | 7/23/19 3:04:00 PM |
|---|---|---|

| Page 15: [21] Formatted | Lucia Deaconu | 7/23/19 3:04:00 PM |
|---|---|---|

Font color: Auto, Not Superscript/ Subscript

| Page 15: [21] Formatted | Lucia Deaconu | 7/23/19 3:04:00 PM |
|---|---|---|

Font color: Auto, Not Superscript/ Subscript

| Page 15: [21] Formatted | Lucia Deaconu | 7/23/19 3:04:00 PM |
|---|---|---|

Font color: Auto, Not Superscript/ Subscript

| Page 15: [21] Formatted | Lucia Deaconu | 7/23/19 3:04:00 PM |
|---|---|---|

Font color: Auto, Not Superscript/ Subscript

| Page 15: [21] Formatted | Lucia Deaconu | 7/23/19 3:04:00 PM |
|---|---|---|

Font color: Auto, Not Superscript/ Subscript

| Page 15: [21] Formatted | Lucia Deaconu | 7/23/19 3:04:00 PM |
|---|---|---|

Font color: Auto, Not Superscript/ Subscript

| Page 15: [21] Formatted | Lucia Deaconu | 7/23/19 3:04:00 PM |
|---|---|---|

Font color: Auto, Not Superscript/ Subscript

| Page 15: [21] Formatted | Lucia Deaconu | 7/23/19 3:04:00 PM |
|---|---|---|

Font color: Auto, Not Superscript/ Subscript

| Page 15: [21] Formatted | Lucia Deaconu | 7/23/19 3:04:00 PM |
|---|---|---|

Font color: Auto, Not Superscript/ Subscript

| Page 15: [21] Formatted | Lucia Deaconu | 7/23/19 3:04:00 PM |
|---|---|---|

Font color: Auto, Not Superscript/ Subscript

| Page 15: [21] Formatted | Lucia Deaconu | 7/23/19 3:04:00 PM |
|---|---|---|

Font color: Auto, Not Superscript/ Subscript

| Page 15: [21] Formatted | Lucia Deaconu | 7/23/19 3:04:00 PM |
|---|---|---|

Font color: Auto, Not Superscript/ Subscript

| Page 15: [22] Deleted | Lucia Deaconu | 7/23/19 7:38:00 PM |
|---|---|---|

| Page 15: [22] Deleted | Lucia Deaconu | 7/23/19 7:38:00 PM |
|---|---|---|

| Page 15: [23] Deleted | Lucia Deaconu | 7/22/19 12:15:00 PM |
|---|---|---|

| Page 15: [23] Deleted | Lucia Deaconu | 7/22/19 12:15:00 PM |
|---|---|---|

| Page 15: [23] Deleted | Lucia Deaconu | 7/22/19 12:15:00 PM |
|---|---|---|

| Page 15: [24] Formatted | Lucia Deaconu | 7/23/19 3:09:00 PM |
|---|---|---|

Font: Not Italic

| Page 15: [24] Formatted | Lucia Deaconu | 7/23/19 3:09:00 PM |
|---|---|---|

Font: Not Italic

| Page 15: [25] Deleted | Lucia Deaconu | 7/18/19 5:17:00 PM |
|---|---|---|

| Page 15: [25] Deleted | Lucia Deaconu | 7/18/19 5:17:00 PM |
|---|---|---|

| Page 15: [25] Deleted | Lucia Deaconu | 7/18/19 5:17:00 PM |
|---|---|---|

| Page 15: [25] Deleted | Lucia Deaconu | 7/18/19 5:17:00 PM |
|---|---|---|

---

## Author Response (AR2)

**Author's response**

Journal: ACP
Title: Satellite inference of water vapor and aerosol-above-cloud combined effect on radiative budget and cloud top processes in the Southeast Atlantic Ocean
Author(s): Lucia T. Deaconu et al.
MS No.: acp-2019-189

Authors want to thank the Co-Editor, Paquita Zuidema, for her contribution and interactive comments. The answers to specific questions are addressed below, while the modifications made in the manuscript are in red.

1. There is a sentence in the abstract that is difficult to parse: "A detailed analysis of the heating rates shows that the absorbing aerosols are 90 % responsible for warming the ambient air where they reside, with approximately +5.7 K/day, while the accompanying water vapour above clouds has a longwave effect of +4.7 K/day (equivalent to 7% decrease) on the cloud-top cooling." There is no altitude provided, and I have trouble reconciling the positive longwave effect with the 7% decrease - in what? - followed by a reference to cloud-top cooling. Are the parentheses misplaced? Since this sentence is in the abstract, I would suggest revisiting it to clarify further for the reader what they should take away from it.

We changed the text with: ". A detailed analysis of the heating rate profiles shows that within the smoke layer at 4 km, the absorbing aerosols are 90% responsible for warming the ambient air with approximately 5.7 K/day. The accompanying water vapor however, has a longwave effect at distance on the cloud top, reducing its radiative cooling in the first 100 m by approximately 4.7 K/day (equivalent to 7%). We infer that this decreased cloud-top cooling in particular, in addition with the higher humidity above the clouds, might modify the cloud-top entrainment rate and its effect, leading to thicker clouds."

2. At the top of p.12, the last sentence in the top paragraph mentions that fuel moisture can make a significant contribution to the humidity within an aerosol plume. A back-of-the-envelope calculation based on Parmar et al., 2008, will contradict this assessment. I encourage the authors to explore this, as this has ramifications for the modelling of the moisture transport.

Potter (2005) suspected evidence of a contribution and proposed the need to determine how much moisture a fire adds to the air and whether this amount is or is not important. Clements et al. (2006) presented an experimental study, from which he deduced a confirmation of Potter's (2005) argument that water vapor from a wildland or grass fire can significantly modify the atmospheric dynamics. Parmar et al., 2008 conducted 16 combustion experiments of different types of biomass fires. His premise was that water vapor released from biomass burning may have different sources: 1) the production of $H_2O$ by chemical reaction and 2) release of moisture that is not chemically bound to the organic molecules of the fuel. His

results showed that non-bound biomass burning moisture ranges from 33% in dry African hardwood to 220 % in the fresh pine branches with needles. They suggest that fuel moisture can make a significant contribution to the water vapour content of fire plumes, but recognize that their study lacks measurements of water vapor release from biomass burning under field conditions, hence it is not possible to constrain their results and the modelling of moisture transport.

In our article we do not affirm that the water vapor can make "a significant contribution to the humidity within the aerosol plume". We merely state that previous studies show that fresh biomass can indeed release water vapor in the atmosphere. Our data show a clear covariance between the water vapor and the aerosol loading, that might be due to a specific regional pattern of circulation (that we show and discuss; also reference Adebiyi et al, 2015). We cannot disregard the possibility of water vapor and biomass burning aerosols released simultaneously from combustion processes, and the effect of these fire plumes on dynamics and moisture transport. Indeed, however, since we do not have the means to prove this statement, we are only suggesting a hypothesis from where the observed moisture comes from.

For clarification we modified the text as following:
Several studies (Potter, 2005, Clements et al., 2006; Parmar et al., 2008) suggest that depending on the moisture content of fresh biomass, the natural or anthropogenic biomass fires are releasing water vapour in the atmosphere (in addition to organic and black carbon, $CO_2$ and CO (Levine, 1990)), that can influence the atmospheric dynamics, thus moisture transport. Without measurements of water vapour release from biomass burning in field conditions, however, it is difficult to constrain the effect on water vapor transport. It might be important to account for the effect of this accompanying moisture, and to identify the different air circulation patterns that will lead the biomass-burning transportation off coast of South Africa.

3.  Are there any systematic changes in some of your findings, Fig 10 and Fig. 11 in particular, as a function of month (JJA)? Fig. 6 shows that there is a seasonal change happening in the winds during these months, consistent with a strengthening of the temperature gradient between southern Africa and the Congo region (e.g., Adebiyi and Zuidema, 2016). One might or might not expect other relationships to also change and it could inform the LWP-qv discussion. I believe the authors have broken down their study by month already so I'm just looking for a short textual assessment here.

We have indeed broken down our study by month, but haven't looked at every month individually. Unfortunately, we haven't checked the relationship between LWP and absorption AOD for September and October. It was discussed at one point, but because of the limited number of 'low' situations in this period, we considered these data are not sufficiently statistically significant and we haven't pursued it further. It can still be done, but not in the time I have allocated now for this paper.
In order to clarify in the text, we added in the conclusions "Our results confirm previous satellite observations and studies that showed that clouds contain more water and are at slightly lower altitudes when large loads of absorbing aerosols are located above them. Indeed, we observed a significant increase of LWP between *low* and *high* cases, whatever the

meteorological conditions (**Error! Reference source not found.** and **Error! Reference source not found.**). These results are valid for June-August, since we have selected one meteorological regime with a similar number of *high* and *low* situations. We do not ignore the possibility that different results may be found in another period (such as September-October), characterised by different meteorological conditions and aerosol emissions."

4. the abstract mentions "the region we focus on" in the 3rd paragraph. be explicit, also about the time period. It adds interest to your study, as no study to date has focused on that particular region.

Thank you. We modified the text with: During this analysis, we account for the variation in the meteorological conditions over our sample area, by selecting the months associated to one meteorological regime (June-August).